# Transcriptional changes detected in fecal RNA from neonatal dairy calves of different breeds following gastrointestinal disease of varying severity

C. S. McConnel ◉*, G. S. Slanzon ¤◉, L. M. Parrish‡, S. C. Trombetta‡, L. F. Shaw ◉‡, D. A. Moore, W. M. Sischo

Department of Veterinary Clinical Sciences, Field Disease Investigation Unit, Washington State University, Pullman, Washington, United States of America

◉ These authors contributed equally to this work.
¤ Current address: Department of Tropical Plant and Soil Sciences, University of Hawai'i at Mānoa, Honolulu, Hawaii, United States of America
‡ LMP, SCT and LFS also contributed equally to this work.
* cmcconnel@wsu.edu

**Data Availability Statement:** All relevant data are within the paper and its Supporting Information files.

## Abstract

Gastrointestinal (GI) disease is a major health concern in preweaned dairy calves. The objective of this fixed cohort study was to use RNA isolated from preweaned Holstein and Jersey heifer calf feces to study the molecular adaptations to variable clinical GI disease. The study was conducted on a commercial calf ranch in the western U.S. Enrolled calves were assessed twice daily for variations in demeanor, milk intake, and hydration. Fecal consistency scores were recorded at enrollment (day 1), and on the day (day 10) that a fecal sample was collected for differential gene expression (DGE). Calves with diarrhea on either day were classified as having either uncomplicated, localized GI disease (scours), or systemic GI disease (systemic enteritis). Eighty-four calves' fecal RNA was evaluated for DGE, of which 33 calves (n = 20 Holstein; n = 13 Jersey) were consistently healthy. The remaining 51 calves (n = 23 Holstein; n = 28 Jersey) experienced varying severity of GI disease during the sampling window. Genes of interest were related to the inflammatory response (i.e., *IFNG*, *NFKB1*, *NOD2*, *TLR2*, and *TLR4*) and cell membrane or cytoplasmic transport (i.e., *AQP3*, *FABP2*, *KRT8* and *SLC5A1*). Breed-specific findings indicated that *AQP3*, *IFNG*, and *TLR4* were upregulated in Holsteins with systemic enteritis, whereas *KRT8* was downregulated in systemically affected Jerseys. Holsteins did not appear affected by scours aside from a tendency for DGE of toll-like receptors (TLRs) on the day of diarrhea. However, Jersey calves consistently demonstrated a tendency to upregulate *IFNG*, *NFKB1*, and *TLR4* when affected with either scours or systemic enteritis. These findings were more pronounced in systemically affected Jersey calves and were observed as a delayed response to both scours and systemic enteritis. These findings support previous observations suggesting that Holstein calves may be better equipped than Jersey calves to rapidly fight pathogen invasion.

**Funding:** This project was supported by Agriculture and Food Research Initiative Competitive Grant no. 2019-68008-29897 from the USDA National Institute of Food and Agriculture, the USDA National Institute of Food and Agriculture, Animal Health & Disease Research Capacity Grant project 1014680, the WSU CVM Caldwell Endowment, and the American Jersey Cattle Association. There was no additional external funding received for this study. The funders had no role in study design, data collection and analysis, decision to publish, or preparation of the manuscript.

**Competing interests:** The authors have declared that no competing interests exist.

## Introduction

Research into dairy heifer health problems tends to simplify disease into preweaned or post-weaned gastrointestinal (GI) or respiratory disease, and diagnostic accuracy is often hindered by a lack of detail [1–3]. The diagnosis of GI disease in particular has relied on the observation of diarrhea, but we historically have lacked the knowledge and tools to discriminate between GI disease phenotypes [4, 5]. Although blood biomarkers of inflammation, oxidative stress, metabolism and hydration provide additional insight into overall animal health, these analyses do not describe the actual biological adaptations of the GI tract (GIT) during the onset and progression of disease [6]. Furthermore, standardized methods of classification have been lacking such that what little information is available is often unproductive. As a result, inconsistencies in disease data presentation hinder descriptions, comparisons, and investigations into the overall consequences of disease on heifer health and wellbeing [7]. These misspecifications of disease processes can result in inappropriate therapeutic and preventive actions that impact lifetime wellbeing and put the dairy industry's reputation at risk for failing to adequately address preventable losses [8].

Efforts to discriminate dairy calf GI disease phenotypes more fully have included transcriptomic assessments of fecal RNA [6]. GI tract epithelial cell exfoliation is associated with the natural turnover process to preserve tissue structure [9], and the daily shedding of these cells into feces provides a source of RNA to evaluate gene expression [10]. This provides a noninvasive sampling method in place of tissue biopsies or postmortem tissue analysis [11]. Importantly, fecal RNA can be differentiated based on the presence of a polyA tail in mammalian RNA that is absent in bacterial RNA. Consequently, Rosa et al. [6] demonstrated differential expression of genes related to cell membrane transport and inflammation in fecal RNA from neonatal Jersey male calves following mild diarrhea. This suggests that RNA isolated from fecal samples provides a potential tool for the evaluation of physiological changes due to GI disease. However, the inflammatory response observed by Rosa et al. in fecal RNA was delayed compared to the pattern of fecal scores and proinflammatory blood biomarkers. Accordingly, further research of transcriptional alterations is warranted to study the intestinal response to inflammation and its variation over time in neonatal dairy calves of different breeds experiencing diverse manifestations of GI disease [12, 13].

Therefore, the current study evaluated transcriptional changes detected in fecal RNA from neonatal dairy calves experiencing varying severity of GI disease with and without therapeutic interventions. We hypothesized that inflammatory and cellular transport signals would be differentially expressed in response to adaptations related to breeds and disease severity and timing. Thus, the objective of this study was to use RNA isolated from fecal samples to study the molecular adaptations to variable clinical GI disease via the expression of select genes involved in inflammation and cellular transport.

## Materials and methods

### Ethics statement

The research protocol was reviewed and approved by the Institutional Animal Care and Use Committee of Washington State University (ASAF#6414).

### Study design and calf enrollment

This fixed cohort study was conducted on a commercial calf ranch in the western U.S. between July 13 and August 6, 2019. The ranch housed approximately 25,000 Holstein, Jersey, and crossbred calves through 200 days of age sourced from multiple dairies. Breed differences were

variable within the source dairies ranging from Holstein and Jersey, Holstein or Jersey only, or crossbred from a single source. Calves were fed colostrum at the dairy of birth, transited <2 hours to the ranch, and arrived at the ranch at ≤1 day of age. Transfer of passive immunity was assessed via blood samples obtained from 1-day old calves via jugular venipuncture to measure total serum protein (TSP) using a calibrated refractometer [14].

## Animal husbandry

On-farm personnel were responsible for all primary care of the calves including feeding, cleaning, watering, and bedding maintenance. The calves were housed in adjacent individual hutches with no direct contact between calves. Upon entry to the ranch all calves were dehorned using a dehorning paste (Dr. Naylor Dehorning Paste, H.W. Naylor Co. Inc., Morris NY, USA), sprayed with a fly control spray (Ultra-Boss Pour-on Insecticide, Intervet Inc., Merck Animal Health, Omaha NE, USA), and ear notched and tested for bovine viral diarrhea virus (BVDV) using PCR. An enteric clostridial vaccine (Ultrabac CD, Zoetis Inc., Kalamazoo MI, USA) and a vitamin complex with selenium and Vitamin E (MU-SE, Intervet Inc., Merck Animal Health, Omaha NE, USA) also were administered on arrival. An intranasal viral respiratory vaccine (Vista Once SQ, Intervet Inc., Merck Animal Health, Omaha NE, USA) was administered at 2 days of age and a booster was administered at 20 days of age. A 7-way clostridial vaccine (Ultrabac 7, Zoetis Inc., Kalamazoo MI, USA) was administered at 45 days of age. Calves were kept in hutches through approximately 90 days of age.

Calves were fed 2 liters of a custom milk blend in a bottle twice daily from 1 to 15 days of age, and 3 liters of the milk blend thereafter until 52 days of age. From 53 to 60 days of age, calves were fed 2 L once per day and weaned thereafter. The milk blend consisted of pasteurized waste milk together with milk replacer and was targeted for an optimal composition of 13% solids, 22–24% fat, and 28% protein. Under the oversight of a licensed veterinarian following veterinary feed directive guidelines, all calves enrolled in the study and between 5 to 12 days of age received milk medicated with neomycin oxytetracycline per the labeled dose (10 mg/lb body weight/day; Neo-Oxy 100/100 MR, PharmGate, Omaha NE, USA), to combat ongoing GI disease problems. Two liters of customized oral electrolyte solution also were offered after the afternoon feeding during this same risk period from 5 to 12 days of age. Water was available between milk feedings, and a handful of grain mix consisting of pellets, molasses, and whole corn was offered from 3 days of age and steadily increased to approximately 2.25 kg by day 30 with free choice thereafter.

## Clinical health assessments and therapeutic interventions

Two on-farm veterinarians oversaw general calf health management and treatment protocols for the ranch. Diagnoses and treatments were based on input from calf health managers' clinical assessments. Morbidity and mortality records included broad disease diagnoses and associated treatments and were managed on-farm using DairyComp 305 (Valley Agricultural Software, Tulare, CA) with record oversight and health data compilation provided through The HEALTHSUM Syndicate LLC (Sunnyside, WA, USA).

For the purposes of this study, a PhD candidate and WSU veterinarian conducted twice daily clinical evaluations based on a standardized calf health-scoring system [4], prior to and at the time of each a.m. and p.m. milk feeding. Assessments began on the day of arrival through the completion of the follow-up period to 21 days of age. Primary points of consideration included demeanor (bright/alert/responsive vs depressed/dull), milk intake (good appetite; did not finish the milk or did not consume any of what was offered; orogastric intubation), hydration status (mm of ocular recession), and fecal consistency scores (1 = well-formed fecal

samples; 2 = semi-formed fecal samples; 3 = loose fecal samples; 4 = watery fecal samples). Due to inconsistencies in fecal appearance within the calf hutches and an inability to observe each calf defecate at the time of all clinical evaluations, fecal consistency scores only were recorded at the time of fecal sampling. Rectal temperature also was recorded for all calves at the time of sampling utilizing a digital thermometer cleaned with alcohol swabs between calves.

Calves with fecal scores of 3 or 4 were diagnosed with diarrhea and classified as having either uncomplicated, localized GI disease (scours), or systemic GI disease (systemic enteritis). Calves with scours had no signs of clinical disease aside from diarrhea. Calves with systemic enteritis had diarrhea along with a behavioral score ≥1 on a matrix incorporating demeanor, mobility, ear positioning, sucking reflex, hydration status based on ocular recession, appetite, and interest in interaction (Table 1). Calves with systemic enteritis were treated medically by farm personnel based on farm protocols incorporating the following alone or in combination: IV fluid therapy (lactated Ringer's solution), anti-inflammatories (flunixin meglumine or dexamethasone), a commercial yeast product (Celmanax SCP, Arm & Hammer, Church & Dwight Co., Inc., Ewing Township, NJ, USA), a proprietary customized probiotic, kaolin-pectin, or variable IM or SQ antibiotics (ceftiofur crystalline free acid, florfenicol, or oxytetracycline). Calves with fecal scores <3 at the time of sampling were classified as consistently healthy if they did not demonstrate abnormal clinical parameters or behavioral scores during a period of 21 days after arrival or receive any treatment for GI disease, respiratory disease, or other health problems during the sampling period.

### Fecal samples for differential gene expression (DGE)

Fecal samples were collected per rectum between July 17 and August 6, 2019 (Fig 1). Heifer calves were randomly enrolled into the sampling scheme and allocated to an initial sampling day (day 1 enrollment sample) between 4 to 12 days of age to have fecal samples (~25–250 g) collected after the morning feeding for microbiome analysis used in a concurrent study [15]. This provided the opportunity to sample healthy calves and calves with previously undiagnosed scours but no other clinical abnormalities. Calves that demonstrated systemic enteritis (i.e., diarrhea plus a behavioral score ≥1) during this same period but prior to a random sample were sampled only once at the time of initial clinical diagnosis irrespective of the randomized sampling scheme.

Nine days after the initial fecal collection each calf was sampled again (day 10 DGE sample; 13–21 days of age) in an effort to align the fecal RNA transcriptomic analysis with previously documented delays in gene expression following evidence of diarrhea [6]. Calves that were initially sampled when healthy or with scours but progressed to systemic enteritis by 12 days of age, were sampled again on the day that systemic enteritis was diagnosed. These calves then

**Table 1. Behavioral assessment scoring matrix for preweaned Holstein and Jersey calves.**

| Score | Demeanor | Mobility | Ears | Sucking Reflex | Ocular recession | Appetite | Interaction |
|-------|----------|----------|------|----------------|------------------|----------|-------------|
| 0 | Bright, alert, responsive | Actively mobile | Erect | Strong | None noted | Robust; consumes all milk on offer | Interactive |
| 1 | Mildly depressed, dull, less responsive | Capable of standing with minimal encouragement | Slightly drooped | Slightly diminished | Slightly sunken; <3 mm | Slow to drink; ≤25% of milk remains | Sluggish |
| 2 | Moderately depressed, dull, less responsive | Capable of standing with moderate encouragement | Drooped | Markedly diminished | Moderately sunken; 3–4 mm | Slow to drink; ≤50% of milk remains | Minimal interest; weak |
| 3 | Markedly depressed, markedly unresponsive | Capable of standing with assistance | Drooped; limp | No suckle; lethargic | Markedly sunken; 5–6 mm | Slow to drink; ≤75% of milk remains | Uninterested; lethargic |
| 4 | Unresponsive | Incapable of standing; sternal or lateral recumbency | Drooped; inert | No suckle; inert | Markedly sunken; >6 mm | Inappetant | Incapable of interaction |

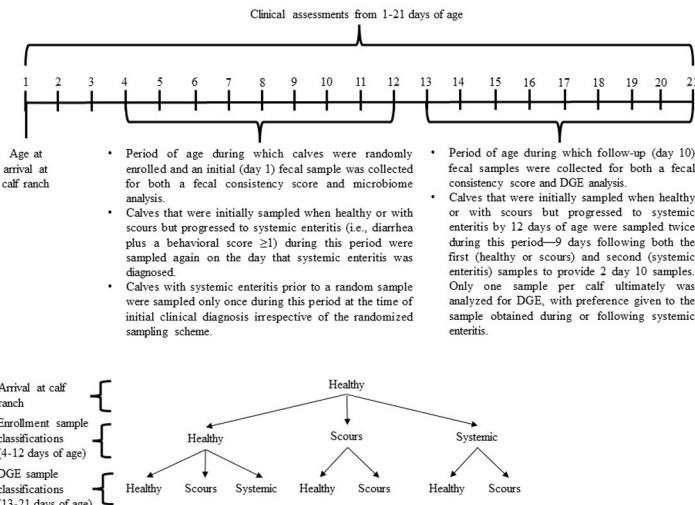

**Fig 1. Timeline for fecal collections and fecal consistency scores, and observed progression of health and disease states included in this analysis.**

were sampled twice more 9 days following both the first (healthy or scours) and second (systemic enteritis) samples to provide 2 day 10 samples. No calf was sampled more than 4 times in total and only one sample ultimately was analyzed for DGE, with preference given to the sample obtained during or following systemic enteritis (e.g., calves with systemic enteritis only were evaluated for disease progression in terms of healthy to systemic, systemic to healthy, or systemic to scours). Fecal samples were collected in sterile Whirl-Pak sampling bags (Cat #14-955-186, Thermo Fisher Scientific, Waltham, MA, USA) and immediately placed in a cooler with ice packs.

## RNA extraction

Fecal samples remained on ice until they were processed in an on-farm laboratory within 2–4 hours of collection. Approximately 0.3 g of feces was weighed into a 15 mL conical tube containing 3 mL of RNA*later* (Cat #AM7021, Thermo Fisher Scientific, Waltham, MA, USA) and then thoroughly mixed by vortex for 30 sec. Fecal bags and RNA*later* tubes containing feces were then frozen onsite at -20˚C. Samples were transported on dry ice to the WSU Field Disease Investigation Unit laboratory and stored at -80˚C for approximately 3 months prior to processing. Fecal dry matter was performed by weighing out 1 g of feces and drying the samples in an incubator at 37˚C for 72 hours. Percent dry matter was calculated by dividing the dry weight by the wet weigh multiplied by 100. Fecal samples in RNA*later* were defrosted on ice and vortexed thoroughly for 30 sec at maximum speed. To aid in the removal of RNA*later* [16], 3 mL of chilled PBS (Cat #AM1840, MagMAX Total Nucleic Acid Isolation Kit, Life Technologies Corp., Austin, TX, USA) was added and each sample was thoroughly mixed by vortex for 30 sec at maximum speed. To remove solids, samples were then centrifuged at 700 g for 1 min at 4˚C. To pellet the cells the supernatant was transferred to a new 15 mL conical tube and centrifuged at maximum speed (4500 g) for 5 min at 4˚C. The supernatant was discarded, and the remaining pellet was resuspended in 235 uL of Lysis Buffer provided by the MagMAX kit. The resuspension then was transferred to a bead tube supplied by the MagMax kit. RNA extraction was carried out beginning at the bead beating step in the MagMAX protocol. 90 uL of RNA Storage Solution (Cat #AM7001, Invitrogen, Carlsbad, CA, USA) was used for the final elution step in place of the MagMAX Elution buffer to improve preservation of

RNA. RNA was quantified using a Qubit 4 Fluorometer (Cat #33226, Invitrogen, Carlsbad, CA, USA). RNA integrity was determined through fragment analysis using a 5200 Fragment Analyzer System (Agilent, Santa Clara, CA, USA), and 60ul of eluted RNA was stored at -80°C until submission for DGE analysis.

Due to the instability of fecal RNA and Nanostring's allowance for DV values as a quality measurement, RNA integrity values were not used for quality assessments. Instead, samples with DV200 (percentage of RNA fragments >200 nucleotides) values >50% and concentrations >20 ng/uL were submitted for diagnostic processing at the Primate Diagnostic Services Laboratory (University of Washington, Seattle, WA, USA). Samples with DV200 values <50% and RNA concentrations <35 ng/uL were re-extracted using frozen feces with a Trizol (Cat #15596018, Invitrogen, Carlsbad, CA, USA) extraction method. An RNA Clean & Concentrator Kit (Cat #R1013, Zymo Research, Irvine, CA, USA) was used on all re-extracted samples, and those re-extracted samples with concentrations >60ng/uL were submitted for diagnostic processing as well.

## Gene selection and gene expression analysis

Nine candidate genes were selected for this study (Table 2). These genes were related to the inflammatory response (i.e., *IFNG*, *NFKB1*, *NOD2*, *TLR2*, and *TLR4*) and cell membrane or cytoplasmic transport (i.e., *AQP3*, *FABP2*, *KRT8* and *SLC5A1*). They were selected based on previous investigations into inflammatory pathologies and intestinal mucosa homeostasis [17–19], and gene expression analysis in fecal RNA from nondiarrheic dairy calves [12] and dairy calves undergoing mild diarrhea [6]. The genes glyceraldehyde 3-phosphate dehydrogenase (*GAPDH*), ribosomal protein S15a (*RPS15A*) and ribosomal protein 9 (RPS9) were used as internal control genes based on their previous identification as suitable controls [6].

Gene expression analysis of the 9 target genes and 3 housekeeping genes was performed using the Nanostring System (Nanostring Technologies, Seattle, Washington, USA), as previously described [20]. Gene expression was measured using a custom TagSet for the selected genes (S1 Table) on the Nanostring nCounter Analysis System (Nanostring Technologies, Seattle, Washington, USA). NanoString technology uses a digital color-coded barcode tag with single-molecule imaging that can detect and count hundreds of unique transcripts per reaction. The geometric mean of the internal control genes was used to normalize the DGE and data retention was accomplished in consultation with NanoString given the novelty of this

**Table 2. Genes examined for differential expression from fecal mRNA obtained from Holstein and Jersey calves.**

| Symbol | Name | Biological function | Biological function and process |
|---|---|---|---|
| *AQP3* | Aquaporin 3 | Cell membrane | Cell membrane with a water channel function—epithelial cell marker |
| *FABP2* | Fatty acid-binding protein 2 | Cytoplasm | Fatty acid-binding protein—epithelial cell marker |
| *IFNG* | Interferon, gamma | Inflammatory response | Important antiviral activity and immunoregulatory functions |
| *KRT8* | Keratin 8 | Cytoplasm | Maintain gut microbiota homeostasis—epithelial cell marker |
| *NFKB1* | Nuclear factor of kappa light polypeptide gene enhancer in B-cells 1 | Inflammatory response | Involved in many biological processes: inflammation, immunity, differentiation, cell growth, tumorigenesis, and apoptosis |
| *NOD2* | Nucleotide-binding oligomerization domain 2 | Inflammatory response | Recognizes bacteria molecules and stimulates an immune reaction |
| *SLC5A1* | Solute carrier family 5 member A1 | Cell membrane | Cell membrane with a sodium-glucose transporter function |
| *TLR2* | Toll-like receptor 2 | Inflammatory response | Participates in the innate immune response to gram positive microbial agents |
| *TLR4* | Toll-like receptor 4 | Inflammatory response | Participates in the innate immune response to gram negative microbial agents |

project's use of fecal mRNA. Briefly, data selection required average raw counts $\geq 4$, $> 5$ genes with expression above background, and $> 20$ for the sum of raw counts for all genes within on sample divided by the scaling factor. The analysis of data was performed using nSolver Analysis Software 4.0, including nCounter Advanced Analysis (version 2.0, Nanostring Technologies, Seattle, Washington, USA). More specifically, fold changes and p-values were calculated using nCounter default settings and a Benjamini-Yekutieli (B-Y) correction for multiple comparisons. The B-Y correction makes the assumption that there may be some biological connection between genes and returns moderately conservative estimates of false discovery rate (FDR). The FDR is the proportion of genes with equal or greater evidence for differential expression (i.e., equal or lower raw p-value) than are expected to be "false discoveries" due to chance.

## Sample size

Previous DGE studies have shown that a minimum of 8 animals should be targeted per group (diseased, healthy), with the sample size based on expectations for a multivariate analysis estimating the log fold-change in a target gene's DGE following a negative binomial distribution [21]. For the purposes of this study the enrollment period was based on the historical prevalence of GI disease of variable severity, mortality rates, average number of calf arrivals per day, and anticipated challenges associated with fecal RNA isolation. Expectations were for quality RNA samples from a minimum of 8 Holstein and 8 Jersey calves within each health status category (consistently healthy, scours, systemic enteritis).

## Results

### Calves and clinical outcomes

A total of 183 heifer calves were enrolled with RNA extracted from fecal samples and submitted for diagnostic processing at the Primate Diagnostic Services Laboratory using the Nanostring System. Average fecal dry weight percentages for consistently healthy, scouring, or systemically affected calves were 26.4 ± 6.2%, 12.5 ± 5.7%, and 13.9 ± 5.9%, respectively. A total of 99 samples (n = 38 consistently healthy; n = 47 scours; n = 14 systemic enteritis) were excluded from analysis primarily due to insufficient fecal mRNA raw counts or expression above background. Details related to the 84 Holstein and Jersey heifer calves remaining in the analysis are presented in Table 3. Those calves were sourced from 22 separate dairies and all calves were BVDV negative, the average TSP was 6.5 ±0.8 g/dL, and the average ages at enrollment and DGE samples were 7.7 ±2.5 days and 16.7 ±2.5 days, respectively.

A total of 33 calves (n = 20 Holstein; n = 13 Jersey) were consistently healthy. The remaining 51 calves (n = 23 Holstein; n = 28 Jersey) experienced varying severity of GI disease at some point during the sampling window. Average TSP was the same for healthy (6.5 ±0.9 g/dL) and diseased calves (6.5 ±0.8 g/dL); however, the average TSP for Jerseys (6.9 ±0.7 g/dL) was greater than for Holsteins (6.0 ±0.7 g/dL; P < 0.0001). Twenty-three of the diseased calves (n = 8 Holstein; n = 15 Jersey) were scouring at the time of their enrollment sample but healthy 9 days later when sampled for DGE. Two Jersey calves were scouring at both sampling times. Thirteen calves (n = 9 Holstein; n = 4 Jersey) were healthy when enrolled but scouring by the time of their DGE sample.

Thirteen of the 51 calves with GI disease demonstrated systemic enteritis during the sampling period (Table 4). All 13 were treated with flunixin and 8 were treated with antibiotics during the sampling period (Table 3). Seven of the systemic enteritis calves (n = 9 Holstein; n = 4 Jersey) were systemically ill when enrolled (average behavioral score = 8; range 2–17) but progressed to healthy status at least 7 days prior to their DGE sample (systemic to healthy).

**Table 3. Details related to enrolled Holstein and Jersey calves.** Systemic behavioral scores (Table 1) and treatments (flunixin and antibiotics) were assigned or administered during the 10-day period from enrollment to DGE sampling.

| Calf ID | Breed[1] | TSP (g/dL)[2] | Enrollment age (days)[3] | Enrollment sample[4-6] | DGE age (days)[7] | DGE sample[4-6] | Flunixin (Y/N) | Antibiotic (drug/N) |
|---|---|---|---|---|---|---|---|---|
| 1377 | H | 6.1 | 10 | Healthy | 19 | Healthy | N | N |
| 1382 | H | 4.7 | 10 | Healthy | 19 | Healthy | N | N |
| 2154 | H | 8.5 | 4 | Healthy | 13 | Healthy | N | N |
| 2258 | H | 5.7 | 7 | Healthy | 16 | Healthy | N | N |
| 2312 | H | 7.0 | 11 | Healthy | 20 | Healthy | N | N |
| 2788 | H | 5.0 | 4 | Healthy | 13 | Healthy | N | N |
| 2812 | H | 6.4 | 5 | Healthy | 14 | Healthy | N | N |
| 3044 | H | 5.3 | 10 | Healthy | 19 | Healthy | N | N |
| 3647 | H | 6.2 | 11 | Healthy | 20 | Healthy | N | N |
| 6980 | H | 6.2 | 8 | Healthy | 17 | Healthy | N | N |
| 6985 | H | 6.8 | 6 | Healthy | 15 | Healthy | N | N |
| 7738 | H | 6.6 | 8 | Healthy | 17 | Healthy | N | N |
| 9542 | H | 6.0 | 7 | Healthy | 16 | Healthy | N | N |
| 10981 | H | 5.6 | 4 | Healthy | 13 | Healthy | N | N |
| 15507 | H | 5.8 | 8 | Healthy | 17 | Healthy | N | N |
| 29043 | H | 5.3 | 9 | Healthy | 18 | Healthy | N | N |
| 29055 | H | 6.1 | 4 | Healthy | 13 | Healthy | N | N |
| 44807 | H | 6.0 | 6 | Healthy | 15 | Healthy | N | N |
| 44808 | H | 6.0 | 5 | Healthy | 14 | Healthy | N | N |
| 90027 | H | 5.6 | 7 | Healthy | 16 | Healthy | N | N |
| 5311 | J | 7.9 | 6 | Healthy | 15 | Healthy | N | N |
| 92977 | J | 7.4 | 9 | Healthy | 18 | Healthy | N | N |
| 93003 | J | 7.5 | 6 | Healthy | 15 | Healthy | N | N |
| 93022 | J | 8.0 | 11 | Healthy | 20 | Healthy | N | N |
| 93024 | J | 7.3 | 4 | Healthy | 13 | Healthy | N | N |
| 93042 | J | 6.6 | 9 | Healthy | 18 | Healthy | N | N |
| 93060 | J | 6.1 | 7 | Healthy | 16 | Healthy | N | N |
| 93077 | J | 6.6 | 4 | Healthy | 13 | Healthy | N | N |
| 93078 | J | 6.9 | 5 | Healthy | 14 | Healthy | N | N |
| 93098 | J | 6.8 | 10 | Healthy | 19 | Healthy | N | N |
| 93120 | J | 7.1 | 8 | Healthy | 17 | Healthy | N | N |
| 93135 | J | 6.4 | 10 | Healthy | 19 | Healthy | N | N |
| 122663 | J | 7.6 | 6 | Healthy | 15 | Healthy | N | N |
| 6981 | H | 5.0 | 7 | Scours | 16 | Healthy | N | N |
| 16663 | H | 6.0 | 10 | Scours | 19 | Healthy | N | N |
| 23328 | H | 5.7 | 5 | Scours | 14 | Healthy | N | N |
| 44805 | H | 6.5 | 6 | Scours | 15 | Healthy | N | N |
| 44821 | H | 5.6 | 11 | Scours | 20 | Healthy | N | N |
| 90032 | H | 7.3 | 7 | Scours | 16 | Healthy | N | N |
| 93031 | H | 6.5 | 5 | Scours | 14 | Healthy | N | N |
| 94001 | H | 6.8 | 12 | Scours | 21 | Healthy | N | N |
| 92965 | J | 6.5 | 8 | Scours | 17 | Healthy | N | N |
| 92972 | J | 6.9 | 7 | Scours | 16 | Healthy | N | N |
| 92988 | J | 7.9 | 9 | Scours | 18 | Healthy | N | N |
| 93025 | J | 7.1 | 9 | Scours | 18 | Healthy | N | N |
| 93051 | J | 6.6 | 10 | Scours | 19 | Healthy | N | N |

*(Continued)*

**Table 3.** (Continued)

| Calf ID | Breed[1] | TSP (g/dL)[2] | Enrollment age (days)[3] | Enrollment sample[4-6] | DGE age (days)[7] | DGE sample[4-6] | Flunixin (Y/N) | Antibiotic (drug/N) |
|---------|----------|---------------|--------------------------|------------------------|-------------------|-----------------|----------------|----------------------|
| 93066 | J | 7.2 | 9 | Scours | 18 | Healthy | N | N |
| 93069 | J | 6.4 | 4 | Scours | 13 | Healthy | N | N |
| 93080 | J | 6.8 | 8 | Scours | 17 | Healthy | N | N |
| 93081 | J | 7.1 | 9 | Scours | 18 | Healthy | N | N |
| 93082 | J | 7.2 | 9 | Scours | 18 | Healthy | N | N |
| 93100 | J | 7.1 | 8 | Scours | 17 | Healthy | N | N |
| 93103 | J | 6.8 | 9 | Scours | 18 | Healthy | N | N |
| 93108 | J | 6.0 | 6 | Scours | 15 | Healthy | N | N |
| 93122 | J | 7.5 | 7 | Scours | 16 | Healthy | N | N |
| 122699 | J | 6.6 | 8 | Scours | 17 | Healthy | N | N |
| 93096 | J | 6.8 | 6 | Scours | 15 | Scours | N | N |
| 93131 | J | 7.7 | 9 | Scours | 18 | Scours | N | N |
| 2806 | H | 5.7 | 12 | Healthy | 21 | Scours | N | N |
| 3648 | H | 5.6 | 4 | Healthy | 13 | Scours | N | N |
| 5171 | H | 6.5 | 6 | Healthy | 15 | Scours | N | N |
| 7185 | H | 5.5 | 5 | Healthy | 14 | Scours | N | N |
| 23324 | H | 6.1 | 10 | Healthy | 19 | Scours | N | N |
| 23334 | H | 6.3 | 5 | Healthy | 14 | Scours | N | N |
| 29052 | H | 5.5 | 10 | Healthy | 19 | Scours | N | N |
| 29070 | H | 6.0 | 8 | Healthy | 17 | Scours | N | N |
| 93138 | H | 6.0 | 4 | Healthy | 13 | Scours | N | N |
| 92998 | J | 7.0 | 4 | Healthy | 13 | Scours | N | N |
| 93010 | J | 5.7 | 4 | Healthy | 13 | Scours | N | N |
| 93014 | J | 6.4 | 8 | Healthy | 17 | Scours | N | N |
| 93035 | J | 7.0 | 6 | Healthy | 15 | Scours | N | N |
| 25217 | H | 5.3 | 10 | Systemic | 19 | Healthy | Y | N |
| 90029 | H | 6.3 | 10 | Systemic | 19 | Healthy | Y | ceftiofur |
| 93132 | H | 6.4 | 12 | Systemic | 21 | Healthy | Y | ceftiofur |
| 93020 | J | 7.9 | 12 | Systemic | 21 | Healthy | Y | ceftiofur |
| 93054 | J | 7.8 | 10 | Systemic | 19 | Healthy | Y | N |
| 93056 | J | 7.2 | 8 | Systemic | 17 | Healthy | Y | N |
| 93105 | J | 7.1 | 12 | Systemic | 21 | Healthy | Y | ceftiofur |
| 2799 | H | 5.3 | 9 | Systemic | 18 | Scours | Y | florfenicol |
| 5312 | J | 7.4 | 12 | Systemic | 21 | Scours | Y | ceftiofur |
| 23332 | H | 5.3 | 4 | Healthy | 13 | Systemic | Y | ceftiofur |
| 23338 | H | 5.4 | 4 | Healthy | 13 | Systemic | Y | N |
| 2063 | J | 4.7 | 11 | Healthy | 20 | Systemic | Y | N |
| 93087 | J | 6.0 | 11 | Healthy | 20 | Systemic | Y | ceftiofur |

[1]Breed: H = Holstein; J = Jersey

[2]TSP: Total serum protein

[3]Enrollment age: Age at which calves were enrolled into this study and fecal samples collected for microbiome analysis [15].

[4]Healthy: calves with fecal scores <3 at the time of sampling without abnormal clinical parameters or behavioral scores indicative of GI disease, respiratory disease, or other health problems.

[5]Scours: calves with fecal scores of ≥3 and classified as having uncomplicated, localized GI disease based on a behavioral score of 0.

[6]Systemic: calves with fecal scores ≥3 and systemic GI disease (systemic enteritis) as evidenced by a behavioral score ≥1.

[7]DGE age: Age at which fecal samples were collected for RNA extraction.

**Table 4. Timing and duration of clinical signs for calves with systemic GI disease.** Behavioral scores are based on the behavioral assessment scoring matrix presented in Table 1.

| Calf ID | Breed[1] | Disease progression[2–4] | Initial age with systemic signs (days) | Rectal temp (˚C) on day of initial systemic signs | Behavioral score on day of initial systemic signs | Final age with systemic signs (days) | DGE age (days)[5] | Time between systemic signs and DGE sample (days) |
|---|---|---|---|---|---|---|---|---|
| 25217 | H | systemic to healthy | 10 | 38.4 | 5 | 11 | 19 | 8 |
| 90029 | H | systemic to healthy | 10 | 37.5 | 15 | 12 | 19 | 7 |
| 93132 | H | systemic to healthy | 12 | 39.2 | 3 | 13 | 21 | 8 |
| 93020 | J | systemic to healthy | 12 | 38.8 | 17 | 12 | 21 | 9 |
| 93054 | J | systemic to healthy | 10 | 37.2 | 12 | 11 | 19 | 8 |
| 93056 | J | systemic to healthy | 8 | 39.2 | 3 | 8 | 17 | 9 |
| 93105 | J | systemic to healthy | 12 | 38.9 | 2 | 12 | 21 | 9 |
| 2799 | H | systemic to scours | 9 | 39.5 | 14 | 9 | 18 | 9 |
| 5312 | J | systemic to scours | 12 | 37.9 | 8 | 12 | 21 | 9 |
| 23332 | H | healthy to systemic | 12 | 38.4 | 10 | 18 | 13 | 0 |
| 23338 | H | healthy to systemic | 12 | 38.1 | 7 | 15 | 13 | 0 |
| 2063 | J | healthy to systemic | 20 | 40.1 | 12 | 20 | 20 | 0 |
| 93087 | J | healthy to systemic | 20 | 36.0 | 18 | 20 | 20 | 0 |

[1]Breed: H = Holstein; J = Jersey

[2]Healthy: calves with fecal scores <3 at the time of sampling without abnormal clinical parameters or behavioral scores indicative of GI disease, respiratory disease, or other health problems.

[3]Scours: calves with fecal scores of ≥3 and classified as having uncomplicated, localized GI disease based on a behavioral score of 0.

[4]Systemic: calves with fecal scores ≥3 and systemic GI disease (systemic enteritis) as evidenced by a behavioral score ≥1.

[5]DGE age: Age at which fecal samples were collected for RNA extraction.

Two calves (n = 1 Holstein; n = 1 Jersey) were systemically ill when enrolled (behavioral scores 14 and 8, respectively) and scouring 9 days later when sampled for DGE (systemic to scours). Four calves (n = 2 Holstein; n = 2 Jersey) were healthy when enrolled but progressed to systemic enteritis (average behavioral score = 12; range 7–18) when sampled for DGE 9 days later (healthy to systemic).

## Differential gene expression

In agreement with a study by Rosa and Osorio [12], raw counts reflecting mRNA abundance in fecal RNA were 8-fold greater on average for *KRT8* than the other endogenous genes of interest (S2 Table). *NOD2* was not expressed in numerous samples and was removed from further analysis. Log$_2$ fold-changes (ratios) in gene expression were evaluated for diseased calves versus those that were consistently healthy. DGE was evaluated overall and by breed for calves with any GI disease (Table 5), scours or systemic enteritis specifically (Table 6), and scours or systemic enteritis based on the day of diagnosis (Tables 7 and 8 and Figs 2 and 3). These sequential analyses were performed in an effort to assess and highlight the importance of discriminating between breed-specific GI disease phenotypes when evaluating the consequences of disease on calf health and wellbeing. Estimates of false discovery rate using the B-Y correction indicated that *TLR4* (B-Y P = 0.01), *AQP3* (B-Y P = 0.03), and *IFNG* (B-Y P = 0.06) were expressed differentially in calves with GI disease at any point during the sampling period (Table 5). It is worth acknowledging that less conservative uncorrected estimates indicated potential differential expression of additional genes including *KRT8* (uncorrected P = 0.03), *FABP2* (uncorrected P = 0.05), and *NFKB1* (uncorrected P = 0.07). When analyzed by breed for calves with GI disease at any point only *TLR4* appeared upregulated in Jerseys (B-Y

**Table 5. Log$_2$ fold-changes for fecal mRNA gene expression in calves with clinical symptoms of GI disease at any point from enrollment until the day 10 differential gene expression (DGE) sample (n = 23 Holstein, n = 28 Jersey) as compared to consistently healthy calves of the same breed(s) (n = 20 Holstein, n = 13 Jersey).**

| Breed | Gene Name | Log$_2$ fold change | Std error (log$_2$) | Lower CL (log$_2$) | Upper CL (log$_2$) | P-value | B-Y p-value[1] |
|---|---|---|---|---|---|---|---|
| Combined (Holstein, Jersey) | AQP3 | 0.89 | 0.27 | 0.36 | 1.42 | 0.0013 | 0.0300 |
| | FABP2 | -0.81 | 0.40 | -1.60 | -0.01 | 0.0464 | 0.2900 |
| | IFNG | 0.85 | 0.30 | 0.26 | 1.44 | 0.0052 | 0.0600 |
| | KRT8 | -0.45 | 0.21 | -0.86 | -0.04 | 0.0317 | 0.2400 |
| | NFKB1 | 0.41 | 0.23 | -0.04 | 0.87 | 0.0729 | 0.3900 |
| | SLC5A1 | 0.44 | 0.29 | -0.12 | 1.01 | 0.1210 | 0.5000 |
| | TLR2 | -0.24 | 0.26 | -0.74 | 0.26 | 0.3399 | 1.0000 |
| | TLR4 | 1.03 | 0.28 | 0.49 | 1.58 | 0.0003 | 0.0100 |
| Holstein | AQP3 | 0.98 | 0.35 | 0.28 | 1.67 | 0.0073 | 0.2700 |
| | FABP2 | -0.56 | 0.56 | -1.64 | 0.54 | 0.3098 | 1.0000 |
| | IFNG | 0.52 | 0.40 | -0.27 | 1.30 | 0.1890 | 1.0000 |
| | KRT8 | -0.38 | 0.29 | -0.94 | 0.19 | 0.1867 | 1.0000 |
| | NFKB1 | -0.01 | 0.32 | -0.64 | 0.61 | 0.9765 | 1.0000 |
| | SLC5A1 | 0.31 | 0.34 | -0.36 | 0.98 | 0.3557 | 1.0000 |
| | TLR2 | -0.81 | 0.36 | -1.51 | -0.10 | 0.0258 | 0.3500 |
| | TLR4 | 0.90 | 0.41 | 0.10 | 1.71 | 0.0285 | 0.3500 |
| Jersey | AQP3 | 0.73 | 0.47 | -0.18 | 1.65 | 0.1155 | 0.5400 |
| | FABP2 | -1.09 | 0.59 | -2.25 | 0.06 | 0.0631 | 0.3400 |
| | IFNG | 1.20 | 0.48 | 0.26 | 2.13 | 0.0131 | 0.1200 |
| | KRT8 | -0.70 | 0.28 | -1.25 | -0.14 | 0.0158 | 0.1200 |
| | NFKB1 | 0.87 | 0.39 | 0.10 | 1.64 | 0.0281 | 0.1700 |
| | SLC5A1 | 0.57 | 0.50 | -0.40 | 1.54 | 0.2382 | 0.8900 |
| | TLR2 | 0.44 | 0.36 | -0.27 | 1.15 | 0.2102 | 0.8700 |
| | TLR4 | 1.16 | 0.36 | 0.45 | 1.87 | 0.0019 | 0.0700 |

[1]B-Y p-value: Benjamini-Yekutieli correction for multiple comparisons.

P = 0.07). However, uncorrected estimates suggested the potential for differential expression of *AQP3*, *TLR2*, and *TLR4* in Holsteins, and *FABP2*, *IFNG*, *KRT8*, and *NFKB1* in Jerseys (uncorrected P < 0.1; Table 5).

A more refined analysis based on disease severity indicated that systemic enteritis impacted fecal mRNA DGE more substantially than uncomplicated scours (Table 6). *AQP3*, *IFNG*, and *TLR4* were upregulated in calves with systemic enteritis (B-Y P ≤ 0.001), and *KRT8* appeared downregulated (B-Y P = 0.08). Breed-specific log$_2$ fold-changes indicated that systemically affected Holsteins (n = 6) upregulated *AQP3* (1.97 ± 0.90; B-Y P = 0.02), *IFNG* (1.98 ± 1.00; B-Y P = 0.02), and *TLR4* (1.57 ± 1.05; B-Y P = 0.09), whereas Jerseys with systemic enteritis (n = 7) downregulated *KRT8* (-0.83 ± -0.50; B-Y P = 0.03). However, uncorrected estimates suggested that systemically affected Jerseys also upregulated *AQP3*, *IFNG*, *TLR4* along with *NFKB1* (uncorrected P < 0.1). Although there were no differences noted in calves with scours using the B-Y correction, uncorrected estimates suggested a similar pattern of DGE in Jerseys with scours (n = 21) as compared to those with systemic enteritis. Scouring Jerseys appeared to upregulate *IFNG*, *TLR4* and *NFKB1* and downregulate *KRT8* (uncorrected P < 0.1). On the other hand, Holsteins with scours (n = 17) did not appear to be similarly affected and only demonstrated evidence of *TLR2* downregulation (uncorrected P = 0.02).

When calves with scours were analyzed based on the day of diagnosis (Table 7 and Figs 2 and 3), only *TLR2* was differentially expressed in Holsteins that were healthy at enrollment but

**Table 6.** Log$_2$ fold-changes for fecal mRNA gene expression in calves with scours (n = 17 Holstein, n = 21 Jersey) or systemic enteritis (n = 6 Holstein, n = 7 Jersey) at any point from enrollment until the day 10 differential gene expression (DGE) sample as compared to consistently healthy calves of the same breed(s) (n = 20 Holstein, n = 13 Jersey).

| Breed | GI disease severity[1,2] | Gene Name | Log$_2$ fold change | Std error (log$_2$) | Lower CL (log$_2$) | Upper CL (log$_2$) | P-value | B-Y p-value[3] |
|---|---|---|---|---|---|---|---|---|
| Combined (Holstein, Jersey) | Scours | AQP3 | 0.61 | 0.29 | 0.04 | 1.18 | 0.0341 | 0.6400 |
| | | FABP2 | -0.55 | 0.43 | -1.40 | 0.30 | 0.2030 | 0.8400 |
| | | IFNG | 0.50 | 0.31 | -0.12 | 1.11 | 0.1174 | 0.8400 |
| | | KRT8 | -0.31 | 0.22 | -0.76 | 0.12 | 0.1611 | 0.8400 |
| | | NFKB1 | 0.33 | 0.25 | -0.15 | 0.82 | 0.1703 | 0.8400 |
| | | SLC5A1 | 0.57 | 0.31 | -0.06 | 1.18 | 0.0751 | 0.8400 |
| | | TLR2 | -0.37 | 0.28 | -0.92 | 0.19 | 0.1959 | 0.8400 |
| | | TLR4 | 0.78 | 0.30 | 0.19 | 1.38 | 0.0106 | 0.3900 |
| Holstein | Scours | AQP3 | 0.62 | 0.40 | -0.15 | 1.41 | 0.1110 | 1.0000 |
| | | FABP2 | -0.21 | 0.58 | -1.36 | 0.92 | 0.7021 | 1.0000 |
| | | IFNG | 0.00 | 0.39 | -0.76 | 0.76 | 0.9928 | 1.0000 |
| | | KRT8 | -0.12 | 0.27 | -0.67 | 0.40 | 0.6253 | 1.0000 |
| | | NFKB1 | -0.10 | 0.36 | -0.81 | 0.61 | 0.7704 | 1.0000 |
| | | SLC5A1 | 0.26 | 0.37 | -0.47 | 0.99 | 0.4755 | 1.0000 |
| | | TLR2 | -0.97 | 0.40 | -1.79 | -0.19 | 0.0175 | 0.6500 |
| | | TLR4 | 0.67 | 0.46 | -0.23 | 1.58 | 0.1407 | 1.0000 |
| Jersey | Scours | AQP3 | 0.52 | 0.49 | -0.43 | 1.47 | 0.2738 | 1.0000 |
| | | FABP2 | -0.89 | 0.64 | -2.12 | 0.37 | 0.1591 | 0.7400 |
| | | IFNG | 0.97 | 0.51 | -0.03 | 1.96 | 0.0563 | 0.4200 |
| | | KRT8 | -0.65 | 0.34 | -1.32 | 0.01 | 0.0538 | 0.4200 |
| | | NFKB1 | 0.80 | 0.40 | 0.01 | 1.58 | 0.0450 | 0.4200 |
| | | SLC5A1 | 0.82 | 0.53 | -0.22 | 1.85 | 0.1182 | 0.6300 |
| | | TLR2 | 0.36 | 0.40 | -0.43 | 1.14 | 0.3665 | 1.0000 |
| | | TLR4 | 0.91 | 0.39 | 0.14 | 1.68 | 0.0224 | 0.4200 |
| Combined (Holstein, Jersey) | Systemic enteritis | AQP3 | 1.69 | 0.42 | 0.87 | 2.50 | 0.0003 | 0.0010 |
| | | FABP2 | -1.57 | 0.77 | -3.06 | -0.07 | 0.0413 | 0.1900 |
| | | IFNG | 1.89 | 0.42 | 1.08 | 2.70 | 0.0001 | 0.0005 |
| | | KRT8 | -0.86 | 0.34 | -1.51 | -0.19 | 0.0144 | 0.0800 |
| | | NFKB1 | 0.64 | 0.37 | -0.09 | 1.37 | 0.0804 | 0.3300 |
| | | SLC5A1 | 0.11 | 0.39 | -0.67 | 0.88 | 0.7755 | 1.0000 |
| | | TLR2 | 0.11 | 0.42 | -0.69 | 0.93 | 0.7726 | 1.0000 |
| | | TLR4 | 1.75 | 0.42 | 0.93 | 2.58 | 0.0003 | 0.0011 |
| Holstein | Systemic enteritis | AQP3 | 1.97 | 0.46 | 1.07 | 2.87 | 0.0007 | 0.0200 |
| | | FABP2 | -1.52 | 1.21 | -3.84 | 0.86 | 0.1629 | 1.0000 |
| | | IFNG | 1.98 | 0.51 | 0.98 | 2.98 | 0.0011 | 0.0200 |
| | | KRT8 | -1.08 | 0.72 | -2.47 | 0.32 | 0.1058 | 0.7900 |
| | | NFKB1 | 0.25 | 0.62 | -0.97 | 1.47 | 0.6236 | 1.0000 |
| | | SLC5A1 | 0.44 | 0.37 | -0.29 | 1.16 | 0.2165 | 1.0000 |
| | | TLR2 | -0.34 | 0.76 | -1.84 | 1.15 | 0.5902 | 1.0000 |
| | | TLR4 | 1.57 | 0.54 | 0.53 | 2.62 | 0.0070 | 0.0900 |
| Jersey | Systemic enteritis | AQP3 | 1.37 | 0.78 | -0.17 | 2.90 | 0.0746 | 0.3500 |
| | | FABP2 | -1.69 | 1.30 | -4.32 | 0.85 | 0.1576 | 0.6500 |
| | | IFNG | 1.89 | 0.74 | 0.44 | 3.33 | 0.0144 | 0.1100 |
| | | KRT8 | -0.83 | 0.25 | -1.32 | -0.33 | 0.0026 | 0.0300 |
| | | NFKB1 | 1.08 | 0.59 | -0.07 | 2.23 | 0.0633 | 0.3400 |
| | | SLC5A1 | -0.16 | 0.72 | -1.60 | 1.24 | 0.7978 | 1.0000 |

*(Continued)*

**Table 6.** (Continued)

| Breed | GI disease severity[1,2] | Gene Name | Log$_2$ fold change | Std error (log$_2$) | Lower CL (log$_2$) | Upper CL (log$_2$) | P-value | B-Y p-value[3] |
|---|---|---|---|---|---|---|---|---|
| | | *TLR2* | 0.71 | 0.59 | -0.43 | 1.87 | 0.1938 | 0.7200 |
| | | *TLR4* | 1.93 | 0.74 | 0.48 | 3.39 | 0.0154 | 0.1100 |

[1]Scours: calves with fecal scores of ≥3 and classified as having uncomplicated, localized GI disease based on a behavioral score of 0.

[2]Systemic: calves with fecal scores ≥3 and systemic GI disease (systemic enteritis) as evidenced by a behavioral score ≥1.

[3]B-Y p-value: Benjamini-Yekutieli correction for multiple comparisons.

diagnosed with scours on day 10 at the time of the DGE sample (n = 9; -1.54 ± -0.79; B-Y P = 0.03). Uncorrected estimates also suggested an upregulation of *TLR4* in Holsteins diagnosed with scours on day 10 (uncorrected P = 0.07). Conversely, Jerseys that were scouring at enrollment but healthy on day 10 at the time of the DGE sample only appeared to upregulate *TLR4* (n = 15; uncorrected P = 0.04). Similarly, upregulation of *IFNG* and *NFKB1* also was indicated in Jerseys diagnosed with scours on day 1 but healthy on day 10 (uncorrected P < 0.05).

When calves with systemic enteritis were analyzed based on the day of diagnosis (Table 8 and Figs 2 and 3), those that were affected at the time of enrollment but returned to a healthy status ≥7 days prior to the DGE sample (n = 3 Holstein; n = 4 Jersey) upregulated *IFNG* (B-Y P = 0.01), *TLR4* (B-Y P = 0.02), and *AQP3* (B-Y P = 0.03) with particularly notable log$_2$ fold-changes (2.33 ± 0.92, 2.18 ± 1.04, and 1.87 ± 1.08, respectively). In addition, uncorrected estimates also suggested upregulation of *NFKBI* and *SLC5AI* along with downregulation of *FABP2* in those same calves (uncorrected P < 0.1). Although evidence was lacking for overall DGE (B-Y P ≤ 1.0) in the 4 calves that were healthy at the time of enrollment but experienced systemic enteritis on day 10 when sampled for DGE (n = 2 Holstein; n = 2 Jersey), breed-specific heat maps suggested that the 2 Holsteins (Fig 2) might have been more severely affected at that time than the 2 Jerseys (Fig 3).

The various breed-specific differences stimulated further investigation into whether healthy Holstein and Jersey calves might differentially express the genes of interest. Although no genes were differentially expressed based on the B-Y correction, a less conservative estimate (uncorrected p ≤ 0.05) did indicate upregulation of *TLR2* and downregulation of *KRT8* in healthy Holsteins as compared to healthy Jerseys (Table 9).

## Discussion

The findings from this study confirmed our hypothesis that inflammatory and cellular transport signals would be differentially expressed in fecal RNA in response to adaptations related to breeds and GI disease severity and timing. Our results suggested that preweaned dairy calves upregulate the genes *AQP3*, *IFNG*, and *TLR4* when experiencing undifferentiated GI disease (scours or systemic enteritis; Table 5). A more refined look at GI disease severity and timing indicated that the DGE in fecal RNA was due primarily to systemic enteritis (Table 6), with the inflammatory response and cellular transport signals potentially delayed compared to clinical signs (Table 8) even though all systemically affected calves were treated with flunixin and most (9/13) were treated with antibiotics (Table 3). Additionally, breed-specific findings indicated that upregulation of *AQP3*, *IFNG*, and *TLR4* in fecal RNA was driven by systemically affected Holsteins, whereas *KRT8* was downregulated in Jersey calves with systemic enteritis (Table 6). Furthermore, *TLR2* was downregulated in Holstein calves with scours on the day of the DGE fecal sample (Table 7). These findings highlight the importance of discriminating between

**Table 7.** Log$_2$ fold-changes for fecal mRNA gene expression in calves with scours diagnosed on day 1 at enrollment (scours to healthy; n = 8 Holstein, n = 15 Jersey), or at the time of the day 10 differential gene expression (DGE) sample (healthy to scours; n = 9 Holstein, n = 4 Jersey), as compared to consistently healthy calves of the same breed(s) (n = 20 Holstein, n = 13 Jersey).

| Breed | Timing of scours[1] | Gene Name | Log$_2$ fold change | Std error (log$_2$) | Lower CL (log$_2$) | Upper CL (log$_2$) | P-value | B-Y p-value[2] |
|---|---|---|---|---|---|---|---|---|
| Combined (Holstein, Jersey) | Day 1 | AQP3 | 0.59 | 0.27 | 0.07 | 1.12 | 0.0287 | 0.5300 |
| | | FABP2 | -0.36 | 0.49 | -1.32 | 0.60 | 0.4610 | 1.0000 |
| | | IFNG | 0.53 | 0.33 | -0.12 | 1.18 | 0.1040 | 0.7700 |
| | | KRT8 | -0.20 | 0.26 | -0.71 | 0.30 | 0.4173 | 1.0000 |
| | | NFKB1 | 0.66 | 0.28 | 0.10 | 1.21 | 0.0214 | 0.5300 |
| | | SLC5A1 | 0.75 | 0.42 | -0.07 | 1.57 | 0.0720 | 0.6700 |
| | | TLR2 | 0.08 | 0.33 | -0.56 | 0.73 | 0.7811 | 1.0000 |
| | | TLR4 | 0.62 | 0.35 | -0.06 | 1.30 | 0.0704 | 0.6700 |
| Holstein | Day 1 | AQP3 | 0.34 | 0.38 | -0.42 | 1.09 | 0.3452 | 1.0000 |
| | | FABP2 | 0.01 | 0.68 | -1.32 | 1.35 | 0.9780 | 1.0000 |
| | | IFNG | -0.14 | 0.41 | -0.94 | 0.66 | 0.7179 | 1.0000 |
| | | KRT8 | -0.08 | 0.39 | -0.86 | 0.69 | 0.8119 | 1.0000 |
| | | NFKB1 | 0.43 | 0.58 | -0.69 | 1.57 | 0.4039 | 1.0000 |
| | | SLC5A1 | 0.53 | 0.52 | -0.47 | 1.54 | 0.2774 | 1.0000 |
| | | TLR2 | -0.34 | 0.66 | -1.64 | 0.96 | 0.5657 | 1.0000 |
| | | TLR4 | 0.26 | 0.67 | -1.03 | 1.57 | 0.6590 | 1.0000 |
| Jersey | Day 1 | AQP3 | 0.61 | 0.45 | -0.29 | 1.50 | 0.1677 | 0.6900 |
| | | FABP2 | -0.65 | 0.69 | -2.00 | 0.70 | 0.3250 | 1.0000 |
| | | IFNG | 1.00 | 0.49 | 0.03 | 1.96 | 0.0426 | 0.3800 |
| | | KRT8 | -0.53 | 0.35 | -1.18 | 0.15 | 0.1222 | 0.6800 |
| | | NFKB1 | 0.93 | 0.42 | 0.10 | 1.75 | 0.0292 | 0.3800 |
| | | SLC5A1 | 0.88 | 0.64 | -0.38 | 2.14 | 0.1617 | 0.6900 |
| | | TLR2 | 0.62 | 0.42 | -0.18 | 1.44 | 0.1275 | 0.6800 |
| | | TLR4 | 0.86 | 0.41 | 0.06 | 1.66 | 0.0357 | 0.3800 |
| Combined (Holstein, Jersey) | Day 10 | AQP3 | 0.57 | 0.62 | -0.64 | 1.78 | 0.3365 | 1.0000 |
| | | FABP2 | -1.08 | 0.76 | -2.56 | 0.40 | 0.1391 | 1.0000 |
| | | IFNG | 0.21 | 0.57 | -0.89 | 1.32 | 0.6958 | 1.0000 |
| | | KRT8 | -0.37 | 0.29 | -0.94 | 0.20 | 0.1900 | 1.0000 |
| | | NFKB1 | -0.20 | 0.30 | -0.81 | 0.39 | 0.4860 | 1.0000 |
| | | SLC5A1 | 0.26 | 0.33 | -0.38 | 0.91 | 0.4114 | 1.0000 |
| | | TLR2 | -1.14 | 0.32 | -1.79 | -0.53 | 0.0008 | 0.0300 |
| | | TLR4 | 1.08 | 0.53 | 0.04 | 2.12 | 0.0420 | 0.5200 |
| Holstein | Day 10 | AQP3 | 0.88 | 0.68 | -0.45 | 2.21 | 0.1693 | 1.0000 |
| | | FABP2 | -0.42 | 0.85 | -2.06 | 1.24 | 0.5917 | 1.0000 |
| | | IFNG | 0.12 | 0.56 | -0.97 | 1.22 | 0.8004 | 1.0000 |
| | | KRT8 | -0.16 | 0.30 | -0.76 | 0.42 | 0.5677 | 1.0000 |
| | | NFKB1 | -0.58 | 0.39 | -1.32 | 0.18 | 0.1219 | 1.0000 |
| | | SLC5A1 | 0.01 | 0.42 | -0.79 | 0.83 | 0.9587 | 1.0000 |
| | | TLR2 | -1.54 | 0.40 | -2.32 | -0.75 | 0.0008 | 0.0300 |
| | | TLR4 | 1.03 | 0.58 | -0.10 | 2.16 | 0.0702 | 0.8700 |
| Jersey | Day 10 | AQP3 | -0.07 | 2.23 | -4.32 | 4.30 | 0.9624 | 1.0000 |
| | | FABP2 | -2.51 | 2.36 | -6.64 | 2.12 | 0.1734 | 1.0000 |
| | | IFNG | 0.31 | 2.29 | -4.32 | 4.81 | 0.8417 | 1.0000 |
| | | KRT8 | -0.65 | 1.10 | -2.84 | 1.51 | 0.3996 | 1.0000 |
| | | NFKB1 | 0.54 | 0.45 | -0.34 | 1.41 | 0.2085 | 1.0000 |
| | | SLC5A1 | 0.81 | 0.56 | -0.30 | 1.90 | 0.1330 | 1.0000 |

*(Continued)*

**Table 7.** (Continued)

| Breed | Timing of scours[1] | Gene Name | Log$_2$ fold change | Std error (log$_2$) | Lower CL (log$_2$) | Upper CL (log$_2$) | P-value | B-Y p-value[2] |
|---|---|---|---|---|---|---|---|---|
| | | TLR2 | -0.48 | 0.73 | -1.89 | 0.95 | 0.3990 | 1.0000 |
| | | TLR4 | 1.18 | 2.01 | -2.74 | 5.11 | 0.4096 | 1.0000 |

[1]Scours: calves with fecal scores of ≥3 and classified as having uncomplicated, localized GI disease based on a behavioral score of 0.

[2]B-Y p-value: Benjamini-Yekutieli correction for multiple comparisons.

breed-specific GI disease phenotypes when describing, comparing, and investigating the overall consequences of disease on calf health and wellbeing.

None of the other genes evaluated in this study demonstrated DGE associated with GI disease when using the B-Y correction. This included *NOD2* which frequently was left undetected. Unfortunately, the case numbers were small for select DGE assessments of breed-specific impacts of disease severity, timing of disease, treatment effects, and disease resolution or progression. These factors and the inability to track scours duration limited our ability to differentiate expression patterns more robustly. Nonetheless, less conservative estimates of DGE suggested informative breed-specific patterns. Unlike their response to systemic enteritis, Holstein calves did not appear affected by scours aside from evidence of DGE of toll-like receptors (TLRs) on the day of diarrhea (Table 7 and Fig 2). On the other hand, Jersey calves consistently demonstrated a tendency to upregulate *IFNG*, *NFKB1*, and *TLR4* when affected with either scours or systemic enteritis (Tables 5–7). These findings were more pronounced in systemically affected Jersey calves (Table 6) and were observed as a delayed response to both scours and systemic enteritis (Tables 7 and 8 and Fig 3).

**Table 8.** Log$_2$ fold-changes for fecal mRNA gene expression in calves with systemic enteritis ≥7 days prior to the day 10 differential gene expression (DGE) fecal sample (i.e., systemic to healthy; n = 3 Holstein, n = 4 Jersey), or on the day of the DGE fecal sample (i.e., healthy to systemic; n = 2 Holstein, n = 2 Jersey), as compared to consistently healthy calves of the same breeds (n = 20 Holstein, n = 13 Jersey).

| Breed | Timing of systemic enteritis[1] | Gene Name | Log$_2$ fold change | Std error (log$_2$) | Lower CL (log$_2$) | Upper CL (log$_2$) | P-value | B-Y p-value[2] |
|---|---|---|---|---|---|---|---|---|
| Combined (Holstein, Jersey) | Prior | AQP3 | 1.87 | 0.55 | 0.79 | 2.94 | 0.0036 | 0.0300 |
| | | FABP2 | -2.09 | 1.01 | -4.06 | -0.11 | 0.0399 | 0.2600 |
| | | IFNG | 2.33 | 0.47 | 1.41 | 3.25 | 0.0002 | 0.0100 |
| | (7–10 days prior to fecal sample collection) | KRT8 | -0.42 | 0.31 | -1.03 | 0.19 | 0.1562 | 0.5800 |
| | | NFKB1 | 0.90 | 0.57 | -0.22 | 2.01 | 0.0919 | 0.3800 |
| | | SLC5A1 | 0.61 | 0.33 | -0.03 | 1.25 | 0.0620 | 0.3300 |
| | | TLR2 | -0.04 | 0.52 | -1.06 | 0.99 | 0.9351 | 1.0000 |
| | | TLR4 | 2.18 | 0.53 | 1.14 | 3.21 | 0.0010 | 0.0200 |
| Combined (Holstein, Jersey) | Current | AQP3 | 0.94 | 1.24 | -1.51 | 3.38 | 0.2977 | 1.0000 |
| | | FABP2 | -0.23 | 2.44 | -5.06 | 4.55 | 0.8878 | 1.0000 |
| | | IFNG | 1.32 | 1.43 | -1.47 | 4.13 | 0.2217 | 1.0000 |
| | (day of fecal sample collection) | KRT8 | -1.65 | 1.18 | -4.06 | 0.66 | 0.1004 | 1.0000 |
| | | NFKB1 | 0.82 | 0.81 | -0.76 | 2.40 | 0.1772 | 1.0000 |
| | | SLC5A1 | -0.48 | 1.36 | -3.18 | 2.20 | 0.6063 | 1.0000 |
| | | TLR2 | 1.04 | 0.95 | -0.81 | 2.91 | 0.1582 | 1.0000 |
| | | TLR4 | 0.63 | 0.96 | -1.25 | 2.52 | 0.3532 | 1.0000 |

[1]Systemic enteritis: calves with fecal scores ≥3 and systemic GI disease as evidenced by a behavioral score ≥1.

[2]B-Y p-value: Benjamini-Yekutieli correction for multiple comparisons.

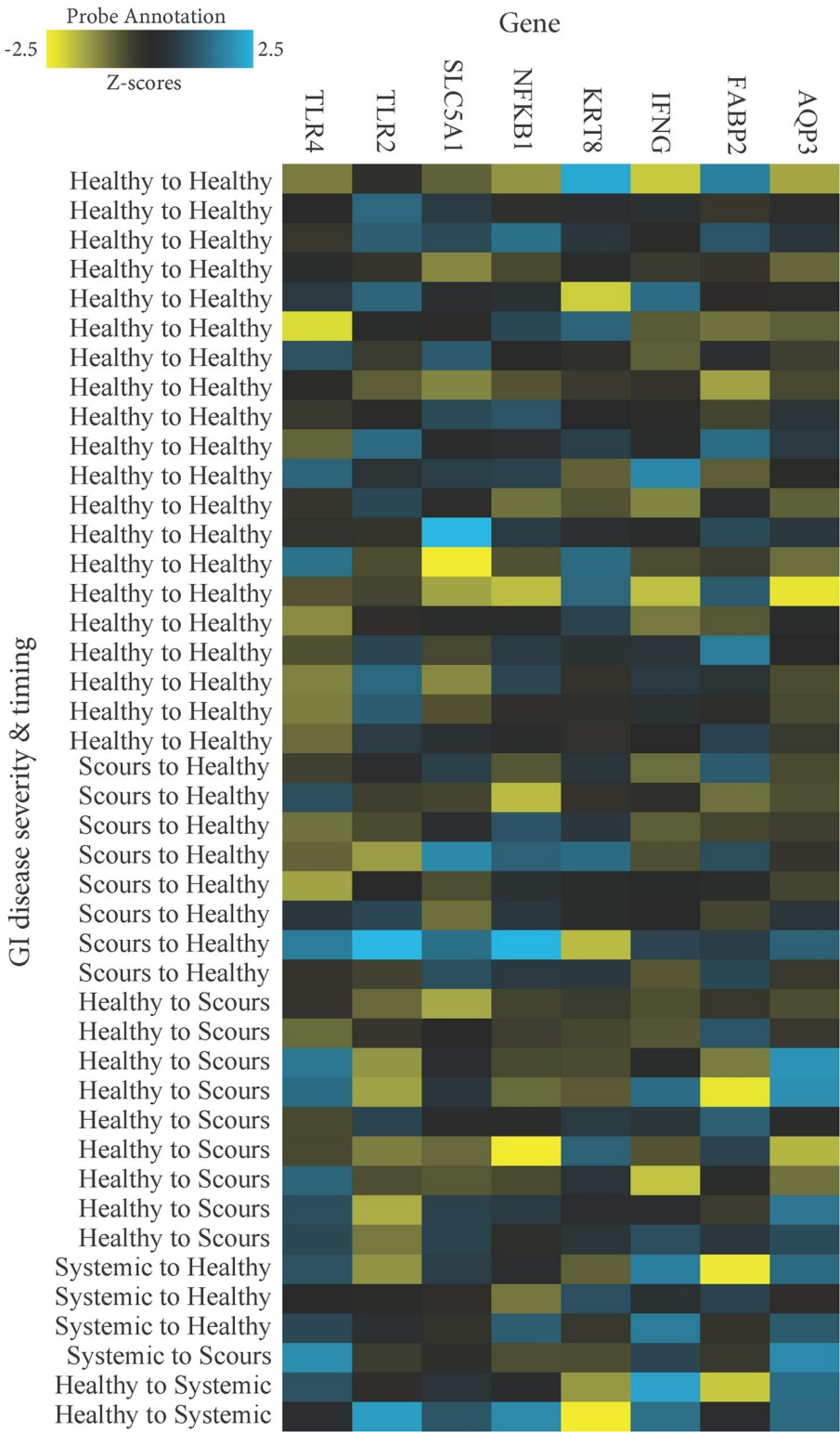

**Fig 2. Heatmap of the normalized data for consistently healthy Holstein calves (healthy to healthy) compared to Holsteins based on the timing of either uncomplicated, localized GI disease (scours to healthy or healthy to scours), or systemic enteritis (systemic to healthy, systemic to scours, or healthy to systemic) scaled to give all genes equal variance, generated via unsupervised clustering.**

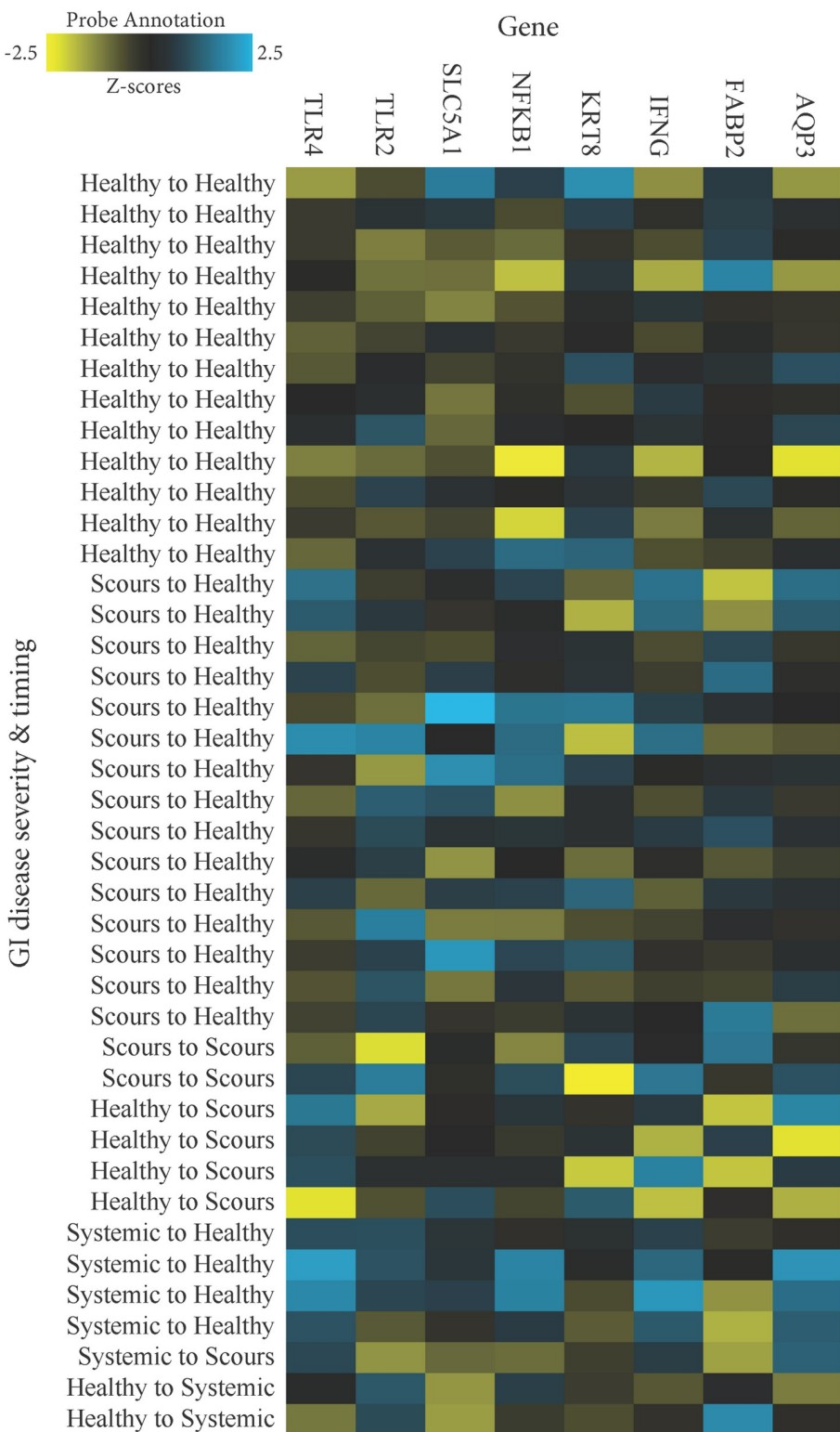

**Fig 3. Heatmap of the normalized data for consistently healthy Jersey calves (healthy to healthy) compared to Jerseys based on the timing of either uncomplicated, localized GI disease (scours to healthy, scours to scours, or healthy to scours), or systemic enteritis (systemic to healthy, systemic to scours, or healthy to systemic) scaled to give all genes equal variance, generated via unsupervised clustering.**

**Table 9. Log$_2$ fold-changes for fecal mRNA gene expression in healthy Holstein (n = 20) vs healthy Jersey (n = 13) calves.**

| Breed | Gene Name | Log$_2$ fold change | Std error (log$_2$) | Lower CL (log$_2$) | Upper CL (log$_2$) | P-value | B-Y p-value[1] |
|---|---|---|---|---|---|---|---|
| Healthy Holsteins compared to healthy Jerseys | AQP3 | -0.29 | 0.43 | -1.15 | 0.57 | 0.4896 | 1.0000 |
| | FABP2 | -0.26 | 0.52 | -1.29 | 0.75 | 0.5976 | 1.0000 |
| | IFNG | 0.28 | 0.46 | -0.62 | 1.18 | 0.5286 | 1.0000 |
| | KRT8 | -0.65 | 0.28 | -1.18 | -0.10 | 0.0221 | 0.2700 |
| | NFKB1 | 0.38 | 0.40 | -0.40 | 1.16 | 0.3240 | 1.0000 |
| | SLC5A1 | 0.04 | 0.48 | -0.89 | 0.99 | 0.9285 | 1.0000 |
| | TLR2 | 0.77 | 0.34 | 0.12 | 1.42 | 0.0219 | 0.2700 |
| | TLR4 | 0.11 | 0.36 | -0.58 | 0.82 | 0.7381 | 1.0000 |

[1]B-Y p-value: Benjamini-Yekutieli correction for multiple comparisons.

These findings substantiate a previous study suggesting that fecal RNA might be used as a noninvasive tool to investigate intestinal transcriptomic profiling of dairy calves [6]. In that study of 8 neonatal Jersey male calves, upregulation of *TLR4* was delayed relative to an increase in fecal scores indicative of mild diarrhea. A similar delay in *TLR4* upregulation appeared to be present in the scouring Jersey calves in our study. On the other hand, the apparent upregulation of *TLR4* in scouring Holstein calves in our study overlapped with both the occurrence of scours and downregulation of *TLR2*. Although this *TLR2* downregulation may have been influenced by the tendency for higher levels of *TLR2* expression in healthy Holstein as compared to Jersey calves (Table 9), a similar downregulation of *TLR2* was demonstrated by Rosa et al. [6] suggesting that much of inflammatory response observed in fecal RNA within that study was due to the upregulation of *TLR4*.

TLRs can detect conserved molecular products of microorganisms and help regulate host adaptive immune responses [22]. Specifically, TLR2 and TLR4 help direct the early innate immune response to microbial challenge [23]. Gram positive lipoteichoic acids (LTA) are recognized by TLR2 whereas TLR4 is responsible for gram negative lipopolysaccharide (LPS) recognition and cell-signaling [24]. Each of these toll-like receptors leads to the expression of distinct inflammatory genes and the induction of various cytokines [23]. For example, an invading pathogen can activate the TLR4 signaling pathway leading to the activation of *NFKB1* which promotes the transcription of genes encoding proinflammatory proteins such as *IL1B*, *TNFA*, *IL6* and *IL8* [25]. This may explain the tendency in our study for concomitant upregulation of both *TLR4* and *NFKB1* in Jersey calves affected with either scours or systemic enteritis. Furthermore, this finding was in agreement with the demonstration by Rosa et al. [6] of *NFKB1* upregulation following a similar delay to that observed for *TLR4*.

In the present study, *IFNG* was upregulated in response to systemic enteritis across breeds but with greater effect in systemic Holsteins than Jerseys. On the other hand, *IFNG* only appeared to be upregulated as a delayed response to scours in Jerseys. Although Rosa et al. [6] demonstrated a time-dependent *IFNG* downregulation in 8 Jersey male calves following mild diarrhea; overall, they showed a pattern of fecal score and pro-inflammatory blood biomarkers (peaking both between 2 and 3 weeks of age) that indicated a pro-inflammatory response during diarrhea in neonatal Jersey calves. It is worth mentioning that a previous study demonstrated that expression of *IL23R* which stimulates interferon gamma (IFNγ) production by T cells was reduced prior to gradual weaning in Jersey relative to Holstein calves [26]. Regardless of potential breed-specific differences, interferons were originally discovered as agents that interfere with viral replications [27], and IFNγ specifically has well-documented antiviral and immunoregulatory properties [28]. Furthermore, IFNγ has been shown to be both an inducer

as well as a regulator of inflammation [29], a duality that confers both resistance and tolerance mechanisms. Although resistance mechanisms are viewed as the primary function of immunity, disease tolerance also is essential in terms of tissue damage control mechanisms that limit the health and fitness costs of infection [30].

The upregulation of *AQP3* in our study was primarily a function of systemic disease in Holsteins although there was evidence of an effect in systemically affected Jerseys and across both breeds with scours. Aquaporin-3 is a cell membrane transporter that transports water, glycerol, urea, and other small uncharged solutes across the plasma membrane [31, 32]. It has been shown to be down-regulated during inflammatory bowel disease in humans [33] and rats [34], and mislocalization of Aquaporin-3 within enterocytes was shown to alter the flux of water through enterocytes in mice with bacterial-induced diarrhea [35]. Furthermore, Rosa et al. [6] demonstrated reduced *AQP3* expression in the fecal RNA from Jersey male calves experiencing mild diarrhea.

As importantly, Rosa and Osorio [12] demonstrated that mRNA expression of *AQP3* tended to be greater in polymorphonuclear leukocytes (PMN) than fecal RNA. Therefore, it is plausible that the fecal RNA samples in our study contained PMN mRNA transcripts and the upregulation of *AQP3* and other genes of interest may have been influenced by PMN infiltration into the intestinal lumen. The transmigration of PMNs into the lumen depends on a variety of activities and signaling events with the goal to maintain homeostasis within the intestine. The various signals that direct such migration include the coordinated efforts of cytokines, adhesion molecules, and highly specific chemokines [36].

The infiltration of immune cells may occur due to altered junction connections and impaired intestinal barrier function [37, 38]. Localized and systemic inflammatory conditions are associated with a leaky epithelial barrier and increased intestinal permeability [39]. A leaky intestinal epithelial barrier allows luminal commensal organism components such as LTA and LPS to enter the submucosa and stimulate the immune system leading to the secretion of regulatory cytokines. The interplay of immune cells, LTA, LPS, and cytokines ultimately augments the inflammatory state of the intestine [40]. In fact, it appears that both LTA and LPS can disrupt barrier function by stimulating increased expression of inducible nitric oxide synthase which leads to increased concentrations of nitric oxide [41]. In addition to disrupting tight junction proteins, nitric oxide is implicated in the induction of enterocyte apoptosis and an accelerated rate of intestinal epithelial cell (IEC) shedding potentially exceeding the rate of IEC production and disturbing intestinal homeostasis [42, 43].

This is relevant to the current study in that Rosa et al. [12, 13] demonstrated that even under nondiarrheic conditions, RNA isolated from neonatal dairy calf feces is derived from a considerable number of IECs as evidenced by mRNA expression of *KRT8*. This held true in our study in that *KRT8* mRNA abundance exceeded that from other genes of interest, suggesting that a considerable amount of fecal RNA originated from IECs. Keratins are the largest family of cellular intermediate filament proteins and are predominant in epithelial cells [44]. Specifically, keratin 8 has been associated with mammary epithelial cells and small intestinal mucosa but not immune cells in general or PMNs in particular [12, 45]. Importantly, various lines of evidence suggest that keratins are key regulators of epithelial integrity [46].

The results from our study suggested a tendency for reduced *KRT8* expression in fecal RNA from healthy Holstein as compared to Jersey calves. Although not noteworthy in every situation, *KRT8* was the only gene demonstrating downregulation in every comparison of diseased versus healthy calves conducted within this study. More specifically, there was a tendency for *KRT8* downregulation in Jerseys with scours and substantial evidence of downregulation in systemically affected Jerseys. Studies conducted in humans and mice suggest that keratin 8 is involved in intestinal barrier protection, colonic active ion transport, colorectal hyperplasia,

and inflammatory regulation [46]. In fact, indications are that keratin 8 may help suppress *NFKB* signaling and TLR-dependent inflammation [47], while protecting colonic integrity in support of gut microbiota homeostasis [46]. This aligns with a previous demonstration of associations between the microbiome and the expression of genes regulating the mucosal barrier and innate immunity in neonatal cattle [48]. The interrelatedness and crosstalk between the various components of the GIT (i.e., epithelium, diverse and robust immune arm, and commensal bacteria) cumulatively make the gut the basis for the health and productivity of animals, highlighting that the GIT is responsible for regulating the physiological homeostasis underlying the ability to withstand infectious and noninfectious stressors [49].

The Holstein and Jersey calves in this study were sourced from different dairies. Consequently, there undoubtedly were undocumented differences in dry cow and maternity management impacting calf health. Nonetheless, Jersey calves demonstrated increased levels of TSP relative to Holsteins suggestive of higher maternally derived serum immunoglobulin G concentrations [50]. Even so, it appeared that despite having received enhanced immunological protection from disease, Jersey calves still may have had reduced immune capacity. Holstein calves experiencing systemic enteritis mounted a more robust and potentially more acute DGE response than systemically affected Jerseys to genes that play key roles in the inflammation and cell membrane or cytoplasmic transport. Furthermore, Jersey calves with GI disease had evidence of *KRT8* downregulation, which may have adversely affected intestinal barrier protection and inflammatory regulation. These findings align with our previous observations that a greater number of cases of GI disease can progress to necrotizing enteritis with peritonitis in preweaned Jersey as compared to Holstein calves (unpublished data). In addition, Holstein as compared to Jersey calves were less affected by scours with fewer delayed DGE effects. These findings support previous observations by Johnston et al. [26] of breed-specific transcriptional activity suggesting Holstein calves may be better equipped than Jersey calves to rapidly fight pathogen invasion before pathogens can cause disease. That study demonstrated differences between these breeds in the expression of genes involved in immune responses and cell signaling activity. More specifically, Jersey calves demonstrated decreased cellular movement, chemotaxis, and phagocytic functionality relative to Holstein calves [26]. Breed-specific cellular effects may be diverse depending on the pathogens and the nature of inflammation which influence the expression of associated resistance and tolerance mechanisms [51]. Ultimately, a timely and properly controlled immune response is the key to an animal's resilience based on balancing pathogen elimination (resistance) and tissue damage (tolerance) [52, 53].

## Conclusions

This study investigated transcriptional changes in fecal RNA from neonatal Holstein and Jersey calves with GI disease of varying severity. Both breeds demonstrated different levels of transcription for important genes that selectively enhance or alter host innate immune defense mechanisms and modulate pathogen-induced inflammatory responses. These findings indicate divergent susceptibilities and immune capacity that provide insight into individual and population-level markers of disease resilience (resistance and tolerance). Although additional research is required to explain functional mechanisms and bioactivity across dairy populations, defining relationships between molecular regulatory features and phenotypic traits may help reduce disease incidence and improve animal welfare. This is particularly relevant to the preweaned period with its metabolic and physiologic changes and associated increase in disease susceptibility. Ultimately, knowledge of breed-specific immune responses may facilitate targeted prophylactic and therapeutic veterinary interventions, guide improvements in host genetic selection, and clarify the burden of disease on animal wellbeing.

## Supporting information

**S1 Table. Probe sequences.**
(XLSX)

**S2 Table. Raw counts for endogenous, housekeeping, negative, and positive probes.**
(XLSX)

## Acknowledgments

The authors thank the participating calf ranch and associated personnel for their invaluable assistance with this project.

## Author Contributions

**Conceptualization:** C. S. McConnel, G. S. Slanzon, D. A. Moore, W. M. Sischo.

**Data curation:** C. S. McConnel, G. S. Slanzon, L. M. Parrish.

**Formal analysis:** C. S. McConnel.

**Funding acquisition:** C. S. McConnel, D. A. Moore, W. M. Sischo.

**Investigation:** C. S. McConnel, G. S. Slanzon, L. M. Parrish, S. C. Trombetta, L. F. Shaw.

**Methodology:** C. S. McConnel, G. S. Slanzon, L. M. Parrish, D. A. Moore, W. M. Sischo.

**Project administration:** C. S. McConnel, G. S. Slanzon.

**Resources:** C. S. McConnel, G. S. Slanzon, L. M. Parrish, S. C. Trombetta, L. F. Shaw.

**Software:** C. S. McConnel.

**Supervision:** C. S. McConnel, G. S. Slanzon.

**Validation:** C. S. McConnel, G. S. Slanzon.

**Visualization:** C. S. McConnel, G. S. Slanzon.

**Writing – original draft:** C. S. McConnel, G. S. Slanzon.

**Writing – review & editing:** L. M. Parrish, S. C. Trombetta, L. F. Shaw, D. A. Moore, W. M. Sischo.

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
