## [Decision Letter · Decision Letter 0]

24 Aug 2022

PONE-D-22-02112Transcriptional changes detected in fecal RNA from neonatal dairy calves of different breeds following gastrointestinal disease of varying severityPLOS ONE

Dear Dr. McConnel,

Thank you for submitting your manuscript to PLOS ONE. After careful consideration, we feel that it has merit but does not fully meet PLOS ONE’s publication criteria as it currently stands. Therefore, we invite you to submit a revised version of the manuscript that addresses the points raised during the review process.

Overall, both reviewers were positive about the manuscript but have made some comments that I am confident you can address through a revision of the manuscript. Apologies for the time that has taken to reach this editorial decision. The final number of invited reviewers was 37!

We look forward to receiving your revised manuscript.

Kind regards,

Angel Abuelo, DVM, MRes, MSc, PhD, DABVP (Dairy), DECBHM

Academic Editor

PLOS ONE

Journal Requirements:

“This project was supported by the WSU CVM Caldwell Endowment (CSM), the American Jersey Cattle Association (CSM), and the USDA National Institute of Food and Agriculture’s Animal Health & Disease Research Capacity Grant project 1014680, and Agriculture and Food Research Initiative Competitive Grant no. 2019-68008-29897 (CSM). The funders had no role in study design, data collection and analysis, decision to publish, or preparation of the manuscript.”

Reviewers' comments:

Reviewer's Responses to Questions

**Comments to the Author**

1. Is the manuscript technically sound, and do the data support the conclusions?

Reviewer #1: Partly

Reviewer #2: Yes

2. Has the statistical analysis been performed appropriately and rigorously? 

Reviewer #1: I Don't Know

Reviewer #2: Yes

3. Have the authors made all data underlying the findings in their manuscript fully available?

Reviewer #1: Yes

Reviewer #2: Yes

4. Is the manuscript presented in an intelligible fashion and written in standard English?

Reviewer #1: Yes

Reviewer #2: Yes

5. Review Comments to the Author

Reviewer #1: The article titled “Transcriptional changes detected in fecal RNA from neonatal dairy calves of different breeds following gastrointestinal disease of varying severity” by C.S. McConnel and collaborators explores an important topic in the dairy industry, focusing the changes in the transcriptome of preweaning calves in response to mild or clinical GI events. The comparison between breeds, as well as the strategy chosen to quantify gene expression, are novel and interesting. However, the manuscript presents several areas that must be improved before it can be deemed suitable for publication. I will list a few broad observations, followed by more precise corrections.

1) Presentation of the experimental design, and statistical soundness: the way the experimental design is explained is confusing. The cohort was housed on a single facility, to which the animals were transported; later, the authors mention that the calves were sourced from different farms (I can’t tell if each breed came from a separate farm, or if every farm provided equal numbers of calves of both breeds). Because of the abstract, I assume all calves were heifers, but the authors do not mention it in the methods. It’s unclear if calf weight and daily grain intake (when offered) were measured, and it appears that the milk intake was only assessed qualitatively. I’m also still unsure as to when the fecal samples were collected: I understand that calves were 4 to 12 days old at collection #1, then were recollected for DEG 10 days later. Except if they were already showing signs of scours, then they were sampled whenever they first presented symptoms + 10 days later? Another reading is: all were sampled initially, then again if they presented scours before d +10, and then twice more (at collection#1 + 10 days and at first symptoms + 10 days). This is very confusing, especially since I can’t understand which sample was ultimately used for DGE in the scours or systemic group, and I have a suspicion that healthy vs non-healthy samples were collected at different time points.

If we account for disease status and time, there should be 5 groups: healthy throughout, healthy to scours, scours to healthy, systemic to healthy and healthy to systemic; other groups, (systemic to scours, scours throughout, systemic throughout) are absent or have too few samples (and they often received different treatment). All of this separated by breed, so 10 groups for pairwise analysis. However, in the presentation of the results, several contrasts are presented, some of which I do not fully understand: what is the comparison in table 6? Is it each non-healthy group vs healthy at DEG? Healthy throughout? In table 8: the comparisons are based on the time of systemic enteritis and are compared to “healthy calves of the same breed”. I assume “prior” means systemic to healthy ; does that mean that they’re compared to all calves that stated out as healthy, regardless of whether they were healthy throughout, or healthy to scours? Or does it imply healthy throughout?

Finally, the groups are also different in size, and I’m not sure if the authors made any adjustments to their statistical approach to account for unequal group sizes. As it stands, a better explanation of sample collection schedule, group assignment and statistical analysis is required. I suggest analyzing based on a combination of disease progression and breed, leading to the aforementioned 10 groups, though I’m not certain that the statistical power will be maintained.

2) Presentation of results: tables are great for a detailed presentation, but virtually impossible to use for cross-group comparisons. I strongly suggest the table be added to the supplementary files, and be replaced with a grouped column graph: the x axis containing the name of the gene of interest, a group of 3 columns per gene (Holstein, jersey, combined), one separate panel per contrast, and some kind of symbol to indicate significance. I’m partial to fold change, rather than log2FC, but both are acceptable; although, please ensure that whatever is chosen is referenced consistently throughout the text (see 242, 311, 326, 333 and other points where it's called “log fold change”). The heat maps are fine, but a per-sample view does easily display group means and is not terribly useful in this case. I suggest to move that to supplementary as well.

3) Reference to microbiome study: the authors refer to a 2022 microbiome study, also published in PLOS, which as I understand was performed on the same samples/animals. The authors should try to combine these findings with those and see if there are any links.

Other notes (with line number)

57: the statement about how information is “wasted” is subjective. Please rephrase

60-61: I’d argue that the welfare of animals is a more relevant implication than the reputation of the dairy industry…

78-79: depending on how the statistical approach is modified, it may not be worth mentioning the therapeutic approach, since it was not treated as a source of variation.

90-95: please specify number of calves enrolled, how many came from which farm, whether they were all female or not.

120: how much grain was offered at the start?

127: missing comma between Tulare and CA

139: can the raw data for rectal temperatures be provided in the supplementary?

147 and elsewhere: please add catalog number for solutions/reagents used

182: homogenized how?

187: “was” should be “were”

188: why the PBS? Also, homogenized how?

189-192: I don’t understand the strategy; first they were vortexed and the solids were removed, then the supernatant was centrifuged again and discarded? Why was this done instead of bead-homogenization and removal of solids, then lysis?

196: report average yield, 260/280 and 260/230 ratios.

198: report mean RIN or equivalent

202: how was this threshold selected?

205-208: were the re-extracted samples quantified for DV200 before submission?

210: I’m unfamiliar with the system, but is it possible to get more detail on probe design?

263: I’m not sure this whole table needs to be here. Maybe a per-group summary of TSP and ages, and then this table could go in supplementary files.

333 and other tables: check consistency on tables. E.g. for table 7, the timing of disease is indicated as day 1 or day 10, while for systemic it’s prior and current. I know they’re not exactly the same in terms of time, but it’s very confusing.

377 and elsewhere: you can use “p-adj” or “adjusted p” instead of “B-Y P”

379 and elsewhere: the fold change here is missing the unit

469-418: the discrepancy between the presented results and the Rosa et al. is not completely addressed. Add a sentence after 481 that ties it together

494: is there evidence in the literature of PMN transmigration in response to scours/diarrhea? Please cite

512 and elsewhere: note a recent follow-up paper by Rosa et al. on the same calves as [6] that looks at expression of the same genes throughout the GI tract using tissue samples https://www.sciencedirect.com/science/article/pii/S0022030220309115.

520: does “not noteworthy” mean “not statistically significant”?

535-554: I’m not sure I agree with the whole argument here. By the logic applied in ~494, KRT8 could indicate larger degree of PMN migration into the lumen, as Rosa and Osorio in ref. 12 noted that KRT8 was downregulated in PMN; at the same time, the study in ref. 25 demonstrates decreased leukocyte migration and chemotaxis. Additionally, could breed differences have been skewed by unequal sample sizes and the contrasts that were chosen? The differences between the two breeds seem minimal (no significant p-adj, only unadjusted p).

565: What markers of disease resilience? Any insight on resilience could be assessed by looking at animals that recovered (scours to healthy and systemic to healthy) vs those that didn’t (scours to scours and systemic to systemic, which are both absent).

Reviewer #2: Dear Authors,

Congratulations on the manuscript. It is well written and easy to understand.

I will list just a few comments:

L21-23 would be "observed by Rosa et al. in fecal..."?

L91 how many animals were previously evaluated in your study? And what are the selection criteria for the animals that you collected fecal samples from?

L91-93 How much colostrum was offered? And what was the concentration of IgG?

L93-95 did the calves have adequate transfer of passive immunity? What is the average value of TSP? Did any calves have failure in transfer of passive immunity?

L113-114 what was the mixture ratio? What is the composition of the liquid diet?

L118 what is the composition of the oral electrolyte solution?

L120-121 what is the composition of the concentrate?

L129-140 what are the references used for such measurements?

6. PLOS authors have the option to publish the peer review history of their article (what does this mean?). If published, this will include your full peer review and any attached files.

Reviewer #1: No

Reviewer #2: No

---

## [Author Response · Author response to Decision Letter 0]

15 Sep 2022

Thank you for your insightful questions and comments. Your commentary has led to changes that we believe make the manuscript a stronger addition to the literature. Please find responses to specific questions below.

Journal Requirements:

“This project was supported by the WSU CVM Caldwell Endowment (CSM), the American Jersey Cattle Association (CSM), and the USDA National Institute of Food and Agriculture’s Animal Health & Disease Research Capacity Grant project 1014680, and Agriculture and Food Research Initiative Competitive Grant no. 2019-68008-29897 (CSM). The funders had no role in study design, data collection and analysis, decision to publish, or preparation of the manuscript.”

AU: Thank you for pointing out our editorial mistakes. We have corrected various file names, amended the Funding Statement and included it within the Cover Letter, and updated the caption for the Supporting Information. 

Review 1 Queries:

The article titled “Transcriptional changes detected in fecal RNA from neonatal dairy calves of different breeds following gastrointestinal disease of varying severity” by C.S. McConnel and collaborators explores an important topic in the dairy industry, focusing the changes in the transcriptome of preweaning calves in response to mild or clinical GI events. The comparison between breeds, as well as the strategy chosen to quantify gene expression, are novel and interesting. However, the manuscript presents several areas that must be improved before it can be deemed suitable for publication. I will list a few broad observations, followed by more precise corrections.

1) Presentation of the experimental design, and statistical soundness: the way the experimental design is explained is confusing. The cohort was housed on a single facility, to which the animals were transported; later, the authors mention that the calves were sourced from different farms (I can’t tell if each breed came from a separate farm, or if every farm provided equal numbers of calves of both breeds). Because of the abstract, I assume all calves were heifers, but the authors do not mention it in the methods. It’s unclear if calf weight and daily grain intake (when offered) were measured, and it appears that the milk intake was only assessed qualitatively. I’m also still unsure as to when the fecal samples were collected: I understand that calves were 4 to 12 days old at collection #1, then were recollected for DEG 10 days later. Except if they were already showing signs of scours, then they were sampled whenever they first presented symptoms + 10 days later? Another reading is: all were sampled initially, then again if they presented scours before d +10, and then twice more (at collection#1 + 10 days and at first symptoms + 10 days). This is very confusing, especially since I can’t understand which sample was ultimately used for DGE in the scours or systemic group, and I have a suspicion that healthy vs non-healthy samples were collected at different time points.

AU: Thank you for identifying points in need of clarification. Calf weights and daily grain intakes were not measured due to logistics and housing constraints on the ranch. Milk intake was assessed qualitatively as stated ((good appetite; did not finish the milk or did not consume any of what was offered; orogastric intubation). We have added clarifying text as follows:

1. Breed differences were variable within the source dairies ranging from Holstein and Jersey, Holstein or Jersey only, or crossbred from a single source.. 

2. Heifer calves were randomly enrolled.

3. The section “Fecal samples for differential gene expression (DGE)” and Fig 1 both have been edited in an attempt to more clearly define the following:

a. Days of age 4-12

i. Period of age during which calves were randomly enrolled and an initial (day 1) fecal sample was collected for both a fecal consistency score and microbiome analysis. 

ii. Calves that were initially sampled when healthy or with scours but progressed to systemic enteritis (i.e., diarrhea plus a behavioral score ≥1) during this period were sampled again on the day that systemic enteritis was diagnosed.

iii. Calves with systemic enteritis prior to a random sample were sampled once during this period at the time of initial clinical diagnosis irrespective of the randomized sampling scheme.

b. Days of age 13-21

i. Period of age during which follow-up (day 10) fecal samples were collected for both a fecal consistency score and DGE analysis.

ii. Calves that were initially sampled when healthy or with scours but progressed to systemic enteritis by 12 days of age were sampled twice during this period—9 days following both the first (healthy or scours) and second (systemic enteritis) samples to provide 2 day 10 samples. 

A key point here is that calves that were randomly enrolled were identified as having scours (or not) when initially sampled whereas systemic enteritis was diagnosed based on clinical presentation. Calves with systemic signs were sampled at the time of diagnosis during the 4-12 days of age window regardless of whether they had already been sampled based on their random allocation. Ultimately one of sample was analyzed for DGE with preference given to the systemic enteritis sample. Contrary to suspicions, healthy and non-healthy samples were collected at the same time.

If we account for disease status and time, there should be 5 groups: healthy throughout, healthy to scours, scours to healthy, systemic to healthy and healthy to systemic; other groups, (systemic to scours, scours throughout, systemic throughout) are absent or have too few samples (and they often received different treatment). All of this separated by breed, so 10 groups for pairwise analysis. However, in the presentation of the results, several contrasts are presented, some of which I do not fully understand: what is the comparison in table 6? Is it each non-healthy group vs healthy at DEG? Healthy throughout? In table 8: the comparisons are based on the time of systemic enteritis and are compared to “healthy calves of the same breed”. I assume “prior” means systemic to healthy ; does that mean that they’re compared to all calves that stated out as healthy, regardless of whether they were healthy throughout, or healthy to scours? Or does it imply healthy throughout?

AU: Indeed there are a number of possible groups—including systemic to scours and scours throughout which are demonstrated in Figs 2 and 3 (no calves survived to populate a group of systemic throughout). To help alleviate confusion, the Table 6 legend has been moved into the title to explain that Table 6 is presenting log fold-changes for fecal mRNA gene expression in calves with scours or systemic enteritis at any point from enrollment until the day 10 DGE sample as compared to healthy calves of the same breed(s).

Thank you for raising the question regarding Table 8, as it highlighted that we should add “consistently” to the descriptor for healthy calves within the titles of Tables 5-8 (i.e., healthy to healthy category), and additional text to Table 6 clarifying categories for comparison (e.g., scours to healthy and healthy to scours). Within the title for Table 8 specifically, we have also included text clarifying that calves with systemic enteritis ≥7 days prior to the day 10 DGE fecal sample were within the “systemic to healthy” groups, whereas calves with systemic enteritis on the day of the DGE fecal sample were within the “healthy to systemic” category.

We discussed how best to present this data given that less useful information was garnered from the collapsed data (i.e., comparing calves with clinical symptoms of GI disease at any point from enrollment until the day 10 DGE) than from the more nuanced analyses of disease severity culminating in the findings presented in Table 8 for systemic disease. In the end, we decided it was important to explore the data in a stepwise fashion in an effort to peel back the layers from broad disease and combined breeds to refined disease and breed-specific outcomes. Although in the grand scheme of things those outer "layers" of the analyses were overly generalized given the final specific outcomes, we felt that omitting those general assessments would lead readers to overlook the importance of the disease classifications. As we stated in the Discussion, these findings highlight the importance of discriminating between breed-specific GI disease phenotypes when describing, comparing, and investigating the overall consequences of disease on calf health and wellbeing. In other words, we are hoping that this approach highlights the fact that scours (diarrhea) should not be considered a singular disease state without consideration for other contextual aspects such as breed and severity.

Finally, the groups are also different in size, and I’m not sure if the authors made any adjustments to their statistical approach to account for unequal group sizes. As it stands, a better explanation of sample collection schedule, group assignment and statistical analysis is required. I suggest analyzing based on a combination of disease progression and breed, leading to the aforementioned 10 groups, though I’m not certain that the statistical power will be maintained.

AU: We truly appreciate your overall insights and identification of issues with clarity. We trust that our edited Fig 1 timeline and additional text throughout has provided the necessary clarity to more easily follow the sampling and analytic sequence. As for adjusting our statistical approach to account for unequal group sizes, nSolver Advanced Analysis runs the R package MASS which accounts for unequal variance when estimating parameters and their variance by estimating a dispersion parameter (*ϕ*) within the negative binomial model. It accommodates the variance of probe expression within biological replicates, which is not of interest in differential expression analysis. When *ϕ* = 0, the negative binomial model reduces to the Poisson model. It is worth noting that the unequal group sizes were a function of both on-farm disease incidence and the mandatory exclusion of 99 samples (n = 38 consistently healthy; n = 47 scours; n = 14 systemic enteritis) due to insufficient fecal mRNA raw counts or expression above background. It was unfortunate that we were unable to use that data, but the reality is that fecal RNA extraction is challenging, and we were glad to have as many useful samples as we did even it did mean that we could not look more closely at select groups lacking a baseline number of samples (i.e., scours to scours, n=2; systemic to scours, n=2). In the end, although we had unequal numbers of animals per group (healthy, scours, systemic disease) and by breed (Jersey, Holstein) we were satisfied that the various comparisons met baseline criteria for sample size and the inclusion of all available data was more useful than restricting data in the interest of equal group sizes.

2) Presentation of results: tables are great for a detailed presentation, but virtually impossible to use for cross-group comparisons. I strongly suggest the table be added to the supplementary files, and be replaced with a grouped column graph: the x axis containing the name of the gene of interest, a group of 3 columns per gene (Holstein, jersey, combined), one separate panel per contrast, and some kind of symbol to indicate significance. I’m partial to fold change, rather than log2FC, but both are acceptable; although, please ensure that whatever is chosen is referenced consistently throughout the text (see 242, 311, 326, 333 and other points where it's called “log fold change”). The heat maps are fine, but a per-sample view does easily display group means and is not terribly useful in this case. I suggest to move that to supplementary as well.

AU: We have edited log fold-change to log2 fold-change where necessary. 

Admittedly, the Tables as presented provide a lot of information through which to sift. However, we respectfully disagree that they should be replaced or added to supplementary files. As we stated above, we think it is important to discriminate between breed-specific GI disease phenotypes and we think that these Tables provide a worthwhile mechanism for doing so. Although simplified versions in the form of columnar graphs might be more visually appealing, they would almost certainly limit our ability to display the full range of data including standard and corrected p-values, which we believe are important for exploring “significant” differences as well as plausible trends in the data. Our goal is to be as transparent as possible with regard to the findings from this study, and we think that presenting the data as-is per comparison is the best (albeit dense) mechanism for doing so. We have proceeded in this manner previously (https://doi.org/10.3389/fvets.2020.559279), and the second Reviewer for this manuscript appears to agree given their comment that this manuscript is easy to understand. 

We will certainly defer to the Editor, but we would also respectfully disagree with moving the heatmaps to the supplementary files. Our experience discussing this data with veterinarians has been that the heat maps provide one of the more powerful representations of the data as it pertains to both individual variations and groups. As such, we would prefer leaving these Figures within the main body of the text as a visual prompt similar to what might be achieved through the addition of columnar graphs. 

3) Reference to microbiome study: the authors refer to a 2022 microbiome study, also published in PLOS, which as I understand was performed on the same samples/animals. The authors should try to combine these findings with those and see if there are any links.

AU: The previously published study was a bit more expansive and involved additional samples taken from throughout the summer of 2019. We considered exploring the overlaps between microbiome and DGE findings but have shelved that idea for the time being due to time and monetary constraints. 

Other notes (with line number)

57: the statement about how information is “wasted” is subjective. Please rephrase

AU: This has been changed to “unproductive”.

60-61: I’d argue that the welfare of animals is a more relevant implication than the reputation of the dairy industry…

AU: Agreed. That is why we say that these misspecifications of disease processes can result in inappropriate therapeutic and preventive actions that impact lifetime wellbeing. Nonetheless, the reputation of the dairy industry is at risk as well and that is an important consideration.

78-79: depending on how the statistical approach is modified, it may not be worth mentioning the therapeutic approach, since it was not treated as a source of variation.

AU: Although not a major component of the analysis, we think that therapeutic interventions should be mentioned simply because they were a component of the management of systemic enteritis.

90-95: please specify number of calves enrolled, how many came from which farm, whether they were all female or not.

AU: These data now are specified within the first paragraph of the Results including reference to the fact that all enrolled calves were female.

120: how much grain was offered at the start?

AU: There was no set amount initially—it was literally a handful. That has been clarified within the text.

127: missing comma between Tulare and CA

AU: Corrected.

139: can the raw data for rectal temperatures be provided in the supplementary?

AU: We decided to only include rectal temperatures in Table 4 to highlight the variability in rectal temperatures even in those calves with systemic evidence of disease. Our experience with rectal temperatures on dairies/calf ranches in the western US in general and with this project specifically is that they are highly variable and ultimately of limited value in clarifying GI disease severity in calves with scours. 

147 and elsewhere: please add catalog number for solutions/reagents used

AU: Solutions such a lactated Ringer’s can be sourced from multiple outlets as can be other of the products used on farm or in the lab. Unless we have overlooked a new PLOS ONE Best Practices in Research Reporting we think that the guidelines mandate that company name and location are the standard practice for reporting solutions and reagents that have a specific origin. 

182: homogenized how?

AU: To avoid any confusion here we restated to say that the samples were thoroughly mixed by vortex.

187: “was” should be “were”

AU: Corrected.

188: why the PBS? Also, homogenized how? 189-192: I don’t understand the strategy; first they were vortexed and the solids were removed, then the supernatant was centrifuged again and discarded? Why was this done instead of bead-homogenization and removal of solids, then lysis?

AU: The PBS and strategy was based on a recommendation from Fisher Scientific’s technical help; we wanted host material rather than everything from the sample. Text has been added to explain and to restate the homogenization.

196: report average yield, 260/280 and 260/230 ratios. 198: report mean RIN or equivalent

AU: Due to the instability of fecal RNA and Nanostring’s allowance for DV values as a quality measurement, RNA integrity values were not used for quality assessments. Instead, samples with DV200 (percentage of RNA fragments >200 nucleotides) values >50% and concentrations >20 ng/uL were submitted for diagnostic processing at the Primate Diagnostic Services Laboratory.

202: how was this threshold selected?

AU: Great question. This was based on discussions with Nanostring’s Field Application Scientist, Dr. Lisa Poole. The short answer is that the Nanostring manual suggests the use of DV200. The longer answer is that there are published instances in the literature that show RIN is not indicative of how successful a gene expression experiment with nCounter will be due to the fact their chemistry does not rely on total RNA integrity that is measured by RIN. Instead, since the barcodes only need about 100bp of intact RNA to bind, it only matters that the RNA is at least 200bps in length, hence the dependence on the DV200 value over RIN.

205-208: were the re-extracted samples quantified for DV200 before submission?

AU: Re-extracted sample DValues were not quantified. Trizol extracts that were equal to or greater then 35ng/ul were allowed to move forward in the process and all re-extracted samples were well above 35ng/ul. In fact, only re-extracted samples with concentrations >60ng/uL were submitted for diagnostic processing per Nanostring recommendations.

210: I’m unfamiliar with the system, but is it possible to get more detail on probe design?

AU: This is probably more than you care to know but the probe design process used by Nanostring breaks a target transcript down into 100nt windows to profile for probe characteristics. Each 100nt window is profiled for intrinsic sequence makeup – non-canonical bases, GC%, inverted and direct repeat regions, runs of polynucleotides, as well as calculating the Tm for each potential probe to target interaction. Each individual probe is then thermodynamically tuned to determine the optimal probe length (each probe ranges in size from 35-50nt) within the 100nt target region. Next, a cross-hybridization score is calculated for each probe region; BLAST is used to identify potential off-target interactions, and then an internal algorithm is used to determine the cross-hybridization score using a pair-wise comparison of the probes to the potential cross-hybridization target. In addition to a cross-hyb score, a splice isoform coverage score is generated by identifying transcripts that are isoforms to the transcript the probe is being designed to. Once all of this information is compiled, then the final probe is selected by identifying the probe that has the optimal splice form coverage, the best cross-hybridization score, and the most optimal thermodynamic profile.

To ensure that there are no potential intramolecular probe-probe interactions that could cause elevated false-positive or background signal for any individual probe in a CodeSet, a stringent intramolecular screen is run on every collection of probes assembled into a CodeSet. This screen is run using a sensitive algorithm that calculates both the Tm and the free energy potential of every probe compared to every other probe in the project. If a probe is identified having either Tm or free energy profile above a conservative threshold, an alternative probe is selected for that target and the screening is re-run until there are no issues.

We considered including some more of this information in the Materials and Methods, but it seems overkill under the circumstances and we felt the inclusion of a relevant reference was more in line with expectations (https://doi.org/10.1186/1756-0500-2-80).

263: I’m not sure this whole table needs to be here. Maybe a per-group summary of TSP and ages, and then this table could go in supplementary files.

AU: This is a bit of a toss-up between providing excessive background information and doing our best to help set the stage for the comparisons to follow. We would prefer leaving this Table as-is even if it is a bit of overkill, to help readers appreciate the enrollment/sampling scheme that is presented in Fig1 while highlighting the various treatments of systemically affected animals. 

333 and other tables: check consistency on tables. E.g. for table 7, the timing of disease is indicated as day 1 or day 10, while for systemic it’s prior and current. I know they’re not exactly the same in terms of time, but it’s very confusing.

AU: Thank you again for pointing out this issue with clarity. We think it has been addressed appropriately per the edits discussed above.

377 and elsewhere: you can use “p-adj” or “adjusted p” instead of “B-Y P”

AU: This is true but we think that B-Y P helps the reader maintain a consistent understanding of where the adjusted p-value originated per the description in the Materials and Methods.

379 and elsewhere: the fold change here is missing the unit

AU: As mentioned above, we have edited log fold-change to log2 fold-change where necessary and included text in the Differential Gene Expression section of the Results indicating that Log2 fold-changes are ratios. 

469-418: the discrepancy between the presented results and the Rosa et al. is not completely addressed. Add a sentence after 481 that ties it together

AU: Thank you very much for highlighting this section. The findings from that paper by Rosa et al. were not well represented in this portion of our manuscript. We should have been pointed out that there was a time-dependent IFNG downregulation in the Jersey calves following mild diarrhea; however, Rosa showed a pattern of fecal score and pro-inflammatory blood biomarkers (peaking both between 2 and 3 weeks of age) that indicated a pro-inflammatory response during diarrhea in neonatal dairy calves. This has been corrected within the text and provides a more egalitarian representation of the comparative data. 

494: is there evidence in the literature of PMN transmigration in response to scours/diarrhea? 

AU: This is a good question. There appears to be little in the way of direct evidence although this paper (https://doi.org/10.1128/IAI.67.9.4950-4954.1999) does suggest that Salmonella-induced enteritis leads to an inflammatory response as evidenced by a rapid and large influx of polymorphonuclear leukocytes into the intestinal mucosa and lumen and a net fluid secretion into the intestinal lumen. Rosa also mentions a good paper by Brazil et al. that speaks to the fact that the migration of PMN across the intestinal epithelium is a common pathological event of many mucosal inflammatory diseases (Brazil et al., 2010). We now have included this reference in our manuscript.

Please cite 512 and elsewhere: note a recent follow-up paper by Rosa et al. on the same calves as [6] that looks at expression of the same genes throughout the GI tract using tissue samples https://www.sciencedirect.com/science/article/pii/S0022030220309115.

AU: Done.

520: does “not noteworthy” mean “not statistically significant”?

AU: More or less—this seemed like an efficient method for pointing out the consistent downregulation without getting mired in B-Y or standard p-value cutoffs. 

535-554: I’m not sure I agree with the whole argument here. By the logic applied in ~494, KRT8 could indicate larger degree of PMN migration into the lumen, as Rosa and Osorio in ref. 12 noted that KRT8 was downregulated in PMN; at the same time, the study in ref. 25 demonstrates decreased leukocyte migration and chemotaxis. Additionally, could breed differences have been skewed by unequal sample sizes and the contrasts that were chosen? The differences between the two breeds seem minimal (no significant p-adj, only unadjusted p).

AU: These are good considerations and worthy of debate. The key from our standpoint is that although Rosa and Osario demonstrated that mRNA expression of AQP3 tended to be greater in PMN than fecal RNA (indicating that the fecal RNA samples in our study may have contained PMN mRNA transcripts), the more important point is that Rosa et al. also demonstrated that even under nondiarrheic conditions RNA isolated from neonatal dairy calf feces is derived from a considerable number of IECs as evidenced by mRNA expression of KRT8. Based on this latter finding we feel comfortable positing that the evidence for KRT8 downregulation in Jersey calves with GI disease may have indicated adversely affected intestinal barrier protection and inflammatory regulation.

As mentioned above, we are confident that nSolver (R package MASS) appropriately accounted for any variance difference and adjusted the estimated model variances for over-dispersed data. That said, we certainly hope that our research and others will continue to explore breed differences in response to GI disease states.

565: What markers of disease resilience? Any insight on resilience could be assessed by looking at animals that recovered (scours to healthy and systemic to healthy) vs those that didn’t (scours to scours and systemic to systemic, which are both absent).

AU: Additional calves representing ongoing disease processes such as systemic-systemic, scours-scours, or systemic-scours would have provided a nice comparative group by which to evaluate potential markers of animal’s resilience based on balancing pathogen elimination (resistance) and tissue damage (tolerance). Hopefully we or others can continue with this line of inquiry to understand varying responses to disease more fully. That said, even without those comparative groups the genes that were selected for analysis based on previous work by Rosa et al. provide insight into resilience based on both breed and disease severity as demonstrated through divergent susceptibilities and apparent immune function.

Review 2 Queries:

Congratulations on the manuscript. It is well written and easy to understand.

AU: Thank you. We appreciate your kind words.

I will list just a few comments:

L21-23 would be "observed by Rosa et al. in fecal..."?

AU: Corrected.

L91 how many animals were previously evaluated in your study? And what are the selection criteria for the animals that you collected fecal samples from?

AU: We trust that our edited Fig 1 timeline and additional text throughout have provided the necessary clarity to more easily follow the sampling and analytic sequence. Specifically, heifer calves were randomly enrolled into the sampling scheme and allocated to an initial sampling day (day 1 enrollment sample) between 4 to 12 days of age to have fecal samples collected. This provided the opportunity to sample healthy calves and calves with previously undiagnosed scours but no other clinical abnormalities. Calves that demonstrated systemic enteritis (i.e., diarrhea plus a behavioral score ≥1) during this same period but prior to a random sample were sampled once at the time of initial clinical diagnosis irrespective of the randomized sampling scheme. Nine days after the initial fecal collection each calf was sampled again (day 10 DGE sample; 13-21 days of age) in an effort to align the fecal RNA transcriptomic analysis with previously documented delays in gene expression following evidence of diarrhea. Calves that were initially sampled when healthy or with scours but progressed to systemic enteritis by 12 days of age, were sampled again on the day that systemic enteritis was diagnosed. These calves then were sampled twice more 9 days following both the first (healthy or scours) and second (systemic enteritis) samples to provide 2 day 10 samples. No calf was sampled more than 4 times in total and only one sample ultimately was analyzed for DGE, with preference given to the sample obtained during systemic enteritis. Ultimately, a total of 183 heifer calves were enrolled with RNA extracted from fecal samples and submitted for diagnostic processing.

L91-93 How much colostrum was offered? And what was the concentration of IgG?

AU: Calves were fed colostrum at the dairy of birth which unfortunately means that there were no records available for specific levels of colostrum that were fed to individual calves. However, transfer of passive immunity was assessed via blood samples obtained from 1-day old calves via jugular venipuncture to measure total serum protein (TSP) using a calibrated refractometer (details below).

L93-95 did the calves have adequate transfer of passive immunity? What is the average value of TSP? Did any calves have failure in transfer of passive immunity?

AU: Average TSP was 6.5 ±0.8 g/dL. The individual TSP levels are presented in Table 3 and interestingly the average TSP was the same for healthy (6.5 ±0.9 g/dL) and diseased calves (6.5 ±0.8 g/dL); however, as expected the average TSP for Jerseys (6.9 ±0.7 g/dL) was greater than for Holsteins 6.0 ±0.7 g/dL; P < 0.0001).

L113-114 what was the mixture ratio? What is the composition of the liquid diet?

AU: The milk blend consisted of pasteurized waste milk together with milk replacer and was targeted for an optimal composition of 13% solids, 22-24% fat, and 28% protein.

L118 what is the composition of the oral electrolyte solution?

AU: The oral electrolytes were a proprietary product for the participating calf ranch, which unfortunately meant that we were not privy to specific compositional details.

L120-121 what is the composition of the concentrate?

AU: Similarly, the concentrate was specifically developed for the calf ranch as a proprietary product so we were provided only with the fact that it consisted of pellets, molasses, and whole corn. 

L129-140 what are the references used for such measurements?

AU: Thank you for this comment as it highlighted the need for an appropriate reference that was missing: McGuirk SM. Disease management of dairy calves and heifers. Vet Clin North Am Food Anim Pract. 2008;24(1):139-53. Epub 2008/02/27. doi: 10.1016/j.cvfa.2007.10.003. PubMed PMID: 18299036.

---

## [Decision Letter · Decision Letter 1]

18 Oct 2022

PONE-D-22-02112R1Transcriptional changes detected in fecal RNA from neonatal dairy calves of different breeds following gastrointestinal disease of varying severityPLOS ONE

Dear Dr. McConnel,

Thank you for submitting your manuscript to PLOS ONE. After careful consideration, we feel that it has merit but does not fully meet PLOS ONE’s publication criteria as it currently stands. Therefore, we invite you to submit a revised version of the manuscript that addresses the points raised during the review process. Please submit your revised manuscript by Dec 02 2022 11:59PM. If you will need more time than this to complete your revisions, please reply to this message or contact the journal office at plosone@plos.org. Please include the following items when submitting your revised manuscript:A rebuttal letter that responds to each point raised by the academic editor and reviewer(s). You should upload this letter as a separate file labeled 'Response to Reviewers'.A marked-up copy of your manuscript that highlights changes made to the original version. You should upload this as a separate file labeled 'Revised Manuscript with Track Changes'.An unmarked version of your revised paper without tracked changes. You should upload this as a separate file labeled 'Manuscript'.

We look forward to receiving your revised manuscript.

Kind regards,

Angel Abuelo, DVM, MRes, MSc, PhD, DABVP (Dairy), DECBHM

Academic Editor

PLOS ONE

Reviewers' comments:

Reviewer's Responses to Questions

**Comments to the Author**

1. If the authors have adequately addressed your comments raised in a previous round of review and you feel that this manuscript is now acceptable for publication, you may indicate that here to bypass the “Comments to the Author” section, enter your conflict of interest statement in the “Confidential to Editor” section, and submit your "Accept" recommendation.

Reviewer #1: (No Response)

2. Is the manuscript technically sound, and do the data support the conclusions?

Reviewer #1: Partly

3. Has the statistical analysis been performed appropriately and rigorously? 

Reviewer #1: I Don't Know

4. Have the authors made all data underlying the findings in their manuscript fully available?

Reviewer #1: Yes

5. Is the manuscript presented in an intelligible fashion and written in standard English?

Reviewer #1: Yes

6. Review Comments to the Author

Reviewer #1: I would like to commend the authors for making pertinent changes to the manuscript, which has improved as a consequence. I still have a few comments to share, which I will list below in no particular order. If line numbers are provided, they refer to the version with track changes.

1) Regarding presentation of timeline and sample collection: I think I now understand how samples were collected. Let me try to lay it out with an example, so that the authors can confirm if I’m correct in my interpretation:

- If a calf is healthy or has scours at enrollment date, and DOES NOT develop systemic enteritis throughout the study � two samples are collected, 9 days apart. Age at collection of sample 1 is 4-12 days, therefore age at sample 2 is 13-21 days. Sample 2 is used for DGE.

- If a calf is healthy or has scours at enrollment date, and DOES develop systemic enteritis throughout the study � sample 1 is collected as above, another sample is collected when symptoms show (let’s call this Sample 1_SE), then sample 2 is collected 9 days after sample 1, and Sample 2_SE is collected 9 days after Sample 1_SE. Ultimately Sample 2_SE is used for DGE.

- If a calf started with systemic enteritis at enrollment date � Sample 1_SE is collected at beginning of symptom, Sample 1 is collected as above, Sample 2 and Sample 2_SE are collected 9 days after corresponding sample. (however, based on Table 3, no animal that started SE had SE at the time the DGE sample was collected).

If the explanation outlined above is correct (and do please correct me if I’m wrong), I have a few comments:

- The explanation of the collection timepoints can be more effectively outlined with something like a flow chart that looks at initial enrollment state, condition throughout the 10 days (healthy, scours, SE), and state at DGE sample. The actual days of age of the calf can be added on a vertical bar next to the graph. In my opinion, the figure that was added in the first round does little to clarify the timeline, as it’s still based on the written description provided with the figure.

- Does table 3 discriminate based on disease progression? For example: if a calf started healthy, quickly developed SE, and was then treated and recovered before the DGE sample, this would show as healthy-healthy, just like a calf that started healthy and ended healthy without change? Maybe there are no such samples? Or does “enrollment sample” in Table 3 refer to “early sample” which would be Sample 1 or Sample 1_SE (see above) depending on whether they developed systemic enteritis or not?

- Since the sample used for DGE are +9 days from sample 1, but sample 1 was collected later for animals that developed systemic enteritis, would this mean that animals that developed systemic enteritis are always slightly older than healthy/scours animals at the DGE sample?

2) Regarding the catalog numbers: both the authors and myself will agree that for common reagents such as lactated Ringer’s, PBS, or certain types of cell culture media adding the catalogue number may be superfluous. However, several nucleic acid extraction and processing kits have been cited within the manuscript, for which many formats (with slightly different chemistries) exist. A few examples: MagMAX Total RNA comes as a 96-preps kit in deep well plates, or as mirVana (still under the “Total RNA” umbrella) which maintains small RNAs. There are four commercial versions of the Qubit fluorimeter available. Invitrogen’s “RNA Storage Solution” (line 205) could be “THE RNA Storage Solution” from Thermo Fisher, or it could be RNA Later which the authors cite multiple times. Whirl Pak bags come in several formats (filtered, unfiltered, in different sizes etc.). The RNA Clean & Concentrator Kit from zymo comes as spin columns, loaded plates or magnetic beads…

Having worked with fecal RNA multiple times in the past, I have first hand experience with how sample collection vessels, storage temperature and solution, and extraction and cleanup can affect quality and integrity of the resulting product. I think it is more than reasonable to request that the author add catalog numbers to these products. The PLOS guidelines are not specific on this, but do request that catalog number be used when antibodies or cell lines are cited in the manuscript, which I think is a rather similar situation to the one we’re discussing here. Please, add the catalog number.

3) Regarding probe design: my question aimed at the possibility of obtaining actual sequences for the probes. If the method used to design them follows the manufacturer’s design and is openly available online, then I will agree that information on the design strategy does not need to be included; the probe sequence, however, would be desirable. One additional note: I could not identify, in the extensive explanation of the probe design provided by the authors, if the probes are designed to span two exons (that is, to straddle an exon-exon junction). If that is not the case, please specify if the TURBO DNAse step of MagMAX was executed or omitted.

4) RNA integrity vs DV200: the requirement for RIN or equivalent in gene expression studies is a debated topic, especially for methods that rely on amplification of short sequences such as RT-qPCR. The crux of the matter is as the authors indicate, amplified sequences are rather short, and as such are less likely to be impacted by degradation. This is true of both qPCR and the nanostring technology employed here, provided the amplicons in qPCR are <200 in length (which they often are). Which is to say, the conversation about RIN requirements is as pertinent to this manuscript as it is to qPCR. The authors mentioned in the response to my feedback that there are “published instances” in the literature where RIN was not indicative of successful quantification. I invite the authors to include a statement about the use of RIN vs DV200 to quantify transcript integrity, citing said manuscripts as support for their strategy. Additionally, the fact that re-extracted samples were not quantified for DV values is a limitation of this study and should be stated or justified, as there can be no guarantee on the quality of those samples, regardless of the method to quantify said quality.

5) The authors indicate that PBS was used to “preserve host material”; is there a reference for this? As considerable or at least some lysis presumably happens before this step (due to vortexing), any lysed pathogenic/microbial nucleic acid material is preserved.

6) Line 193: at what temperature were the samples stored?

7. PLOS authors have the option to publish the peer review history of their article (what does this mean?). If published, this will include your full peer review and any attached files.

Reviewer #1: No

---

## [Author Response · Author response to Decision Letter 1]

18 Nov 2022

Thank you for your additional input. questions and comments. Please find responses to specific questions below.

Reviewer #1: I would like to commend the authors for making pertinent changes to the manuscript, which has improved as a consequence. I still have a few comments to share, which I will list below in no particular order. If line numbers are provided, they refer to the version with track changes.

1) Regarding presentation of timeline and sample collection: I think I now understand how samples were collected. Let me try to lay it out with an example, so that the authors can confirm if I’m correct in my interpretation:

- If a calf is healthy or has scours at enrollment date and DOES NOT develop systemic enteritis throughout the study, two samples are collected, 9 days apart. Age at collection of sample 1 is 4-12 days, therefore age at sample 2 is 13-21 days. Sample 2 is used for DGE.

AU: Correct.

- If a calf is healthy or has scours at enrollment date, and DOES develop systemic enteritis throughout the study, sample 1 is collected as above, another sample is collected when symptoms show (let’s call this Sample 1_SE), then sample 2 is collected 9 days after sample 1, and Sample 2_SE is collected 9 days after Sample 1_SE. Ultimately Sample 2_SE is used for DGE.

AU: Correct.

- If a calf started with systemic enteritis at enrollment date, Sample 1_SE is collected at beginning of symptom, Sample 1 is collected as above, Sample 2 and Sample 2_SE are collected 9 days after corresponding sample. (however, based on Table 3, no animal that started SE had SE at the time the DGE sample was collected).

AU: Calves with systemic enteritis prior to selection for a random sample were sampled only once during this period at the time of initial clinical diagnosis. I.e., there was no Sample 1 collected if a Sample 1_SE preceded the anticipated randomized sample. We have added the word ‘only’ to the text and Fig1 to help clarify this point. 

You are correct that no surviving calves that started SE remained SE 9 days later.

If the explanation outlined above is correct (and do please correct me if I’m wrong), I have a few comments:

- The explanation of the collection timepoints can be more effectively outlined with something like a flow chart that looks at initial enrollment state, condition throughout the 10 days (healthy, scours, SE), and state at DGE sample. The actual days of age of the calf can be added on a vertical bar next to the graph. In my opinion, the figure that was added in the first round does little to clarify the timeline, as it’s still based on the written description provided with the figure.

AU: Figure 1 has been updated to include a flowchart indicating observed enrollment and DGE sample classifications.

- Does table 3 discriminate based on disease progression? For example: if a calf started healthy, quickly developed SE, and was then treated and recovered before the DGE sample, this would show as healthy-healthy, just like a calf that started healthy and ended healthy without change? Maybe there are no such samples? Or does “enrollment sample” in Table 3 refer to “early sample” which would be Sample 1 or Sample 1_SE (see above) depending on whether they developed systemic enteritis or not?

AU: You are correct that ‘enrollment sample’ refers to the sample preceding the DGE sample. E.g., for the purposes of our analysis it was the sample used to enroll a calf into the study. Only one sample ultimately was analyzed for DGE. E.g., no samples from a calf with systemic enteritis were used for anything other than evaluating healthy to systemic, systemic to healthy, or systemic to scours. This might not have been clear given our previous word choice stating that preference was given to the sample obtained during systemic enteritis. This has been corrected within the text to clarify that preference was given to the sample obtained during or following systemic enteritis. 

- Since the sample used for DGE are +9 days from sample 1, but sample 1 was collected later for animals that developed systemic enteritis, would this mean that animals that developed systemic enteritis are always slightly older than healthy/scours animals at the DGE sample?

AU: Correct. Ages were slightly different. Per Table 3, calves that survived following systemic enteritis were initially identified between 8-12 days of age, leading to a DGE sample obtained between 17-21 (average 19.6) days of age. All other DGE samples, including those obtained from calves that were initially healthy but developed systemic enteritis by the time of their DGE sample, were between 13-21 (average 16.4) days of age. Prior to conducting this study, it was decided that the sampling timeframe for DGE between 13-21 days of age would be analyzed in bloc given the relatively narrow spectrum of development. Ongoing research breaking down transcriptomic and DGE changes by day of age may ultimately prove enlightening as we continue to try and understand developmental aspects underlying immunologic resilience (resistance and tolerance). 

2) Regarding the catalog numbers: both the authors and myself will agree that for common reagents such as lactated Ringer’s, PBS, or certain types of cell culture media adding the catalogue number may be superfluous. However, several nucleic acid extraction and processing kits have been cited within the manuscript, for which many formats (with slightly different chemistries) exist. A few examples: MagMAX Total RNA comes as a 96-preps kit in deep well plates, or as mirVana (still under the “Total RNA” umbrella) which maintains small RNAs. There are four commercial versions of the Qubit fluorimeter available. Invitrogen’s “RNA Storage Solution” (line 205) could be “THE RNA Storage Solution” from Thermo Fisher, or it could be RNA Later which the authors cite multiple times. Whirl Pak bags come in several formats (filtered, unfiltered, in different sizes etc.). The RNA Clean & Concentrator Kit from zymo comes as spin columns, loaded plates or magnetic beads…

Having worked with fecal RNA multiple times in the past, I have first hand experience with how sample collection vessels, storage temperature and solution, and extraction and cleanup can affect quality and integrity of the resulting product. I think it is more than reasonable to request that the author add catalog numbers to these products. The PLOS guidelines are not specific on this, but do request that catalog number be used when antibodies or cell lines are cited in the manuscript, which I think is a rather similar situation to the one we’re discussing here. Please, add the catalog number.

AU: Done.

3) Regarding probe design: my question aimed at the possibility of obtaining actual sequences for the probes. If the method used to design them follows the manufacturer’s design and is openly available online, then I will agree that information on the design strategy does not need to be included; the probe sequence, however, would be desirable. One additional note: I could not identify, in the extensive explanation of the probe design provided by the authors, if the probes are designed to span two exons (that is, to straddle an exon-exon junction). If that is not the case, please specify if the TURBO DNAse step of MagMAX was executed or omitted.

AU: An additional Supplementary Table 1 (S1 Table) has been added to provide the actual sequences for the probes. Per Nanostring’s Field Application Scientist, the probes are designed based on transcriptomics data with the introns spliced out. Some may fall across exon-exon junctions, although others may not for targeting the RNA from that gene. So, if the question is how do we know the probe is not targeting DNA, if it is not covering a exon-exon junction unique to RNA? Then the answer is that the probe hybridization assay takes place at 65°C, and the temperature in the PCR block never rises past this so all genomic DNA remains double stranded and inaccessible to the probes, only single-stranded RNA should be binding.

4) RNA integrity vs DV200: the requirement for RIN or equivalent in gene expression studies is a debated topic, especially for methods that rely on amplification of short sequences such as RT-qPCR. The crux of the matter is as the authors indicate, amplified sequences are rather short, and as such are less likely to be impacted by degradation. This is true of both qPCR and the nanostring technology employed here, provided the amplicons in qPCR are <200 in length (which they often are). Which is to say, the conversation about RIN requirements is as pertinent to this manuscript as it is to qPCR. The authors mentioned in the response to my feedback that there are “published instances” in the literature where RIN was not indicative of successful quantification. I invite the authors to include a statement about the use of RIN vs DV200 to quantify transcript integrity, citing said manuscripts as support for their strategy. Additionally, the fact that re-extracted samples were not quantified for DV values is a limitation of this study and should be stated or justified, as there can be no guarantee on the quality of those samples, regardless of the method to quantify said quality.

AU: I think it is important to note that the NanoString technology does not used any “amplified” sequences. There are no enzymatic steps, they are just counting the original number of RNA molecules in the sample. So in this way, the relationship of RIN to qPCR is not the same as RIN to NanoString nCounter assays. Many NanoString applications also use FFPE samples for their RNA, and RIN is not considered a sensitive measure of RNA quality for FFPE samples.

That said, if there is a DV200 less than 50%, Nanostring mostly recommends increasing the sample input. It turns out that according to Nanostring there are MANY (the majority) of groups who do not even check the DV200, they just look at concentration, A260/A280 (1.7-2.3) and A260/A230 (1.8-2.3). They recommend including the DV200 metric if you think there is a reason your samples will be degraded (FFPE). There are many positive and negative controls within in the assay, as well as QC measurements of the data’s signal and quality after running the assay. If there are a lot of QC flags, then it’s possible that something is off and it is recommended to go back and run the sample on a bioanalyzer to double-check the DV200. Otherwise, the assay worked as expected. If the DV200 was poor, we would have expected to get poor signal/background ratios that raise a QC flag.

5) The authors indicate that PBS was used to “preserve host material”; is there a reference for this? As considerable or at least some lysis presumably happens before this step (due to vortexing), any lysed pathogenic/microbial nucleic acid material is preserved.

AU: Correct, during vortexing some mechanical lysis can happen and microbial material will carry over. Of course, that is why we run housekeeping genes to ensure host material was present. Our use of PBS to aid in removal of RNAlater was informed by methods presented in a manuscript entitled, “Variation of glucoraphanin metabolism in vivo and ex vivo by human gut bacteria.” Fei Li et al. PMID: 21342607 PMCID: PMC3137642 DOI: 10.1017/S0007114511000274. This reference has been added to the manuscript.

6) Line 193: at what temperature were the samples stored?

AU: -80°C. This has been added to the text.

---

## [Editor Report · Decision Letter 2]

22 Nov 2022

Transcriptional changes detected in fecal RNA from neonatal dairy calves of different breeds following gastrointestinal disease of varying severity

PONE-D-22-02112R2

Dear Dr. McConnel,

We’re pleased to inform you that your manuscript has been judged scientifically suitable for publication and will be formally accepted for publication once it meets all outstanding technical requirements.

Kind regards,

Angel Abuelo, DVM, MRes, MSc, PhD, DABVP (Dairy), DECBHM

Academic Editor

PLOS ONE
---

## [Editor Report · Acceptance letter]

24 Nov 2022

PONE-D-22-02112R2 

Transcriptional changes detected in fecal RNA from neonatal dairy calves of different breeds following gastrointestinal disease of varying severity 

Dear Dr. McConnel:

I'm pleased to inform you that your manuscript has been deemed suitable for publication in PLOS ONE. Congratulations! Your manuscript is now with our production department. 

Kind regards, 

on behalf of

Dr. Angel Abuelo 

Academic Editor

PLOS ONE